# Extraction, phytochemical characterization and anti-cancer mechanism of Haritaki churna: An ayurvedic formulation

Md Rafi Uz Zama Khan[1], Emiko Yanase[2], Vishal Trivedi[1] *

1 Department of Biosciences and Bioengineering, Indian Institute of Technology-Guwahati, Guwahati, Assam, India, 2 Faculty of Applied Biological Sciences, Gifu University, Gifu, Japan

* vtrivedi@iitg.ernet.in, Vishalash_1999@yahoo.com

**Data Availability Statement:** All relevant data are within the paper and its Supporting Information files.

**Funding:** The author(s) received no specific funding for this work.

## Abstract

Haritaki churna (HC), a single herb ayurvedic formulations is known to be prescribed for various gastro-intestinal disorders in Ayurveda. Haritaki churna aqueous extract (HCAE) has anti-cancer activity against different types of cancer cells with an $IC_{50}$ in the range of 50–97 µg/ml. Bioavailability of Haritaki Churna is very high in digestive track and treatment of colorectal cancer cells HCT-116, DLD1, HT-29 with HCAE reduces its cellular viability with anti-cancer $IC_{50}$ 70µg/ml. HCAE consumption is safe for human as it didn't affect the cellular viability of primary human PBMCs or non-cancerogenic HEK-293 cells. Haritaki churna was found to be stable in biological gastric fluids and bioactive agents are not losing their anti-cancer activity under such harsh conditions. The HPLC Chromatogram of HCAE is giving 13 major peaks and 11 minor peaks. Exploiting LC-MS, IR and NMR spectroscopic techniques, a total of 13 compounds were identified from HCAE namely *Shikimic acid*, *Chebulic acid*, *gallic acid*, *5-hydroxymethylfurfural*, *Protocatechuic acid*, *4-O-galloyl-shikimic Acid*, *5-O-galloyl-shikimic Acid*, *Methylgallate*, *corilagin*, *1, 2, 6, Tri-O-galloyl β-D-glucose*, *chebulagic acid*, *chebulinic acid*, and *Ellagic acid*. Reconstitution and subtraction of phytochemicals from the mixture indicate that E*llagic acid* significantly contribute into anti-cancer effect of HCAE. Cancer cells treated with ellagic acid from HCAE were incapable of completing their cell-cycle and halted the cell-cycle at DNA synthesis S-Phase, as demonstrated by decreased cyclin A2 expression levels with increasing ellagic acid concentration. Halting of cells at S-phase causes induction of apoptosis in cancer cells. Cancer cells exhibiting DNA fragmentation, changes in expression of several apoptotic proteins such as Bcl2, cytochrome-c and formation of cleaved products of caspase 3 and PARP-1 suggests ellagic acid induces cell death via mitochondrial pathway of apoptosis.

## 1 Introduction

'Ayurvedic medicine' or 'ayurveda' known as the traditional medicine of India makes both herbal and metallic formulations using plants, animals and mineral sources as medicines for various ailments [1]. It is practiced successfully in many countries. Many therapeutic products

**Competing interests:** The authors have declared that no competing interests exist.

have been extracted from plants which are mentioned in ayurvedic literature. Ayurvedic formulations are shown to have anti-oxidant [2], anti-diabetic [3], anti-inflammatory [4], anti-asthmatic [5], anti-arthritic [6], anti-pyretic [7], immunomodulatory [8],neuroprotective [9], anti-cancer [10–12],anti-obesity [13], and gastro-protective [14] properties. Triphala, which is made from equal parts of the fruits Haritaki (*Terminalia chebula* Retz.), Bibhitaki (*Terminalia bellerica* Gaertn.), and amla (*Emblica officinalis* Gaertn.), is central to studies on ayurvedic formulations. Triphala has been extensively studied and is shown to have numerous properties [15–19]. Haritaki churna (HC) is a single herb ayurvedic formulation consisting of plant material from *Terminalia chebula* Retz. Haritaki is composed of a variety of phytochemicals such as polyphenols, terpenoids, terpenes, flavonoids, sterols etc. [20]. Haritaki churna is a form of powdered formulation and its plant source has been predominantly used for treatment of gastro-intestinal disorders such as indigestion, gastritis, ulcers, constipation, vomiting, colic pain, gastro-intestinal disorders among other malaises [21]. There have been no studies reported on the efficacy of Haritaki churna to support its claim of being an ayurvedic medicine to treat gastro-intestinal illnesses. With this hindsight, we aimed to investigate the anti-cancer potential on several cancer cell lines such as colorectal cancer (HCT-116, DLD1 and HT-29), cervical cancer (HeLa), breast cancer (MDAMB-231), and osteosarcoma (MG-63) in light of its applications and localized effects on gastro-intestinal related illnesses. We also intended at isolating & characterizing the major chemical compounds present in it and the most active ingredient/s from its rich phytochemical content. This study found that aqueous extract of Haritaki churna is completely capable of killing cancer cells, which can be attributed to the presence of several bioactive compounds, the majority of which are polyphenols. A re-constitution and subtractive analysis of the compounds in the aqueous extract revealed that ellagic acid was the most potent ingredient in the composition. Further, it was also observed that ellagic acid is capable of arresting cell cycle progression in S-phase and induction of mitochondrial pathway of apoptosis in colorectal cancer cells.

## 2 Materials and methods

### 2.1 Chemicals

Haritaki churna was purchased from a local Baidyanath store in Guwahati, Assam, India. Dulbecco's modified eagle's medium, 3-(4, 5-dimethylthiazol-2-yl)-2, 5-diphenyltetrazolium bromide (MTT) from Sigma Aldrich (St. Louis, MO, USA) and Percoll were purchased from Sigma (St. Louis, MO, USA). DMSO (cell-culture grade), Foetal Bovine Serum (FBS), Penicillin-Streptomycin (100X) antibiotic solution, Phosphate Buffer Saline (PBS), Sodium Azide, trypan blue, and trypsin were obtained from HiMedia (Mumbai, India). All the cell culture plates and dishes were purchased from Corning, Lowell, MA, USA. MDAMB-231 (Breast cancer cell line), DLD-1 (Colorectal cancer), HCT-116 (Colorectal cancer), HT-29 (Colorectal cancer), MG-63 (Osteosarcoma), HeLa (Cervical cancer), and HEK-293 (Kidney epithelial) cell lines were procured from National Centre for Cell Sciences, Pune, India.Acetone-$d_6$ (Deuterated acetone), DMSO-$d_6$ (Deuterated DMSO), $CD_3OD$ (Deuterated methanol), and $D_2O$ (Deuterated water) were purchased from Kanto Chemical Co. Inc. Tokyo, Japan. Pancreatin, Sodium taurocholate, lecithin, and maleic acid were procured from HiMedia (Mumbai, India). Glyceryl monooleate and sodium oleate were procured from sigmaaldrich. Apoptosis/ necroptosis antibody sampler kit (cat. # 92570), and antibodies to cyclin A2, GAPDH were purchased from cell signaling technology, Inc (USA). Antibodies detecting β-actin (cat. # BB-AB0024), Bcl2 (cat. # BB-AB0230), PARP-1 (cat. # BB-AB0280) were procured from BioBharathi Life science private ltd (Ind.) Anti-Cytochrome-C (338200) antibody was procured

from Invitrogen. Cyclin D1 (cat. # ab134175) was purchased from Abcam (UK). All other reagents and chemicals were of analytical grade purity.

## 2.2 Sample preparation for cell culture treatments

One gram of Haritaki churna was taken and dissolved in 5 ml of water. The mixture was kept for incubation at 37˚C for 2 hrs at 150 rpm. The mixture was subjected to centrifugation for 10 min at 6000 rpm at room temperature. The procedure was repeated with 5 ml water. The supernatant was stored at 4˚C until further use.

## 2.3 Cell culture

MDAMB-231 cells (Breast cancer cell line), HeLa (Cervical cancer cell line), DLD1, HT29, HCT-116 (Colon cancer cell lines), MG-63 (Osteosarcoma cell line), and HEK-293 (Human embryonic kidney) cells were cultured in DMEM: F12 High glucose media, supplemented with 10% fetal bovine serum (FBS) and 1% penicillin-streptomycin antibiotic solution (100units/ml penicillin and 100μg/ml streptomycin sulfate). Cells were grown in humidified 5% $CO_2$ incubator at 37˚C.

## 2.4 MTT cell viability assay

One day prior to the treatment with HC aqueous extract, cancer cells ($1x10^4$) were seeded in 0.2ml complete medium overnight in 96 well dish. On the day of the experiments, cells were washed twice with cell culture grade phosphate buffer saline (PBS) and subjected to treatment with different concentration of HC aqueous extract (0–1000μg/ml) for 48hrs in serum-free medium. Post treatment, cells were washed with PBS twice and cell viability was measured by MTT assay as described previously [22, 23]. Images of the untreated and treated cells were taken from random 10 fields using the Cytell cell imaging system (GE healthcare).

## 2.5 Preparation of peripheral blood mononuclear cells (PBMCs)

Blood was taken from a healthy volunteer and PBMCs were isolated using HiSeP from HiMedia[TM] [24] as per manufacturer's protocol. The PBMC cells were re-suspended in DMEM-F12 high glucose medium supplemented with 1% Penicillin-Streptomycin antibiotic solution and 10% FBS. The cells were treated with varying concentrations of Haritaki churna aqueous extract and the cell viability of the PMBCs was calculated using MTT cell viability assay.

## 2.6 Preparation of dissolution media

**2.6.1 Simulated gastric fluid (SGF).** Simulated gastric fluid was prepared as described [25]. In brief, 2 grams of sodium chloride is added to 600–700 ml of deionised or distilled water. The pH is then set to 1.2. After setting the pH, 3.2 grams of pepsin (800–1200 units per milligram of activity) is added and stirred until dissolved. The volume of the mixture is made up to 1 litre using deionised or distilled water.

**2.6.2 Fasted state Simulated Intestinal Fluid (FaSSIF).** FaSSIF was prepared as described [25]. FaSSIF was prepared with sodium taurocholate (3 mM), lecithin, (0.2 mM), maleic acid (19.12 mM), sodium chloride (68.62 mM), and sodium hydroxide (34.8 mM) in 1 litre of deionized water with a pH of 6.5.

**2.6.3 Fed state Simulated Intestinal Fluid (FeSSIF).** FeSSIF was prepared as described [25]. FeSSIF was prepared with sodium taurocholate (10 mM), lecithin, (2 mM), glyceryl momooleate (5mM), sodium oleate (0.8 mM), maleic acid (55.02 mM), sodium chloride (125.5 mM), and sodium hydroxide (81.65 mM) in 1 litre of deionized water with a pH of 5.8.

## 2.7 Phytochemical analysis of aqueous extract of Haritaki churna

The presence of phenolics, flavonoids, terpenoids, alkaloids and proteins in the HCAE was qualitatively analyzed using standard detection methods.

### 2.7.1 Total phenolic content (TPC)

The total phenolic content of HCAE was determined using folin-ciocalteau method as described [26]. Gallic acid (GA) was used as standard and the TPC of the extract is expressed in terms of mg GA equivalents/gram dry weight of extract.

**2.7.2 Total flavonoid content (TFC).** The total flavonoid content of HCAE was estimated by the method as described [27]. Quercetin (Q) was used as a standard and the total flavonoid content was expressed in terms of mg Q equivalents/gram dry weight of extract.

**2.7.3 Total alkaloid content (TAC).** The total alkaloid content of HCAE was determined using Dragendorff's test as described [28]. Berberine (BE) was used as a standard and the total alkaloid content was expressed in terms of mg BE equivalents/gram dry weight of extract.

**2.7.4 Total protein content (TPrC).** The total protein content of HCAE was estimated using Bradford assay as described [29]. BSA (Bovine serum albumin) was used as standard and the total protein content was expressed in terms of mg BSA equivalents/gram dry weight of extract.

**2.7.5 Total terpenoid content (TTC).** The total terpenoid content of HCAE was determined using Salkowski's test as described [30]. Enoxolone (EO) was used as a standard and the total terpenoid content was expressed in terms of mg EO equivalents/gram dry weight of extract.

## 2.8 Fractionation of Haritaki churna aqueous extract using reverse-phase open column chromatography

Open column chromatography was performed for the separation of fractions from the supernatant obtained by dissolving the ayurvedic formulation in water as discussed previously. The column is prepared with Diaion HP20SS (Mitsubishi chemical, Japan) resin. The column is filled with HP20SS resin to 20 cm in a 60 cm long column having an inner diameter of 1.5 cm. The column was then conditioned with water equivalent to four volumes of resin length. The sample was poured onto the resin and allowed to settle for few minutes. The column was then eluted serially with increasing ratio of methanol to water as follows: Fraction 1: Methanol-water (0:100), Fraction 2: Methanol-water (20:80), Fraction 3: Methanol-water (50:50), Fraction 4: Methanol-water (80:20), Fraction 5: Methanol-water (100:0). The fractions were dried using rotary evaporator and kept at 4˚C for further use.

## 2.9 Gradient HPLC analysis

The High performance liquid chromatography analysis was used to determine the polyphenols present in the aqueous extract of Haritaki churna. The column used was YMC-TRIART $C_{18}$- (4.6ID X 250mm) and the flow rate was maintained at 1mL/min. The oven temperature was kept constant at 35˚C throughout the HPLC runtime and absorbance was recorded at 254nm. The mobile phases were A: acetonitrile (0.5% HCOOH) and B: water (0.5% HCOOH). The linear gradient of A in B from 0% to 100% over 30 minutes.

## 2.10 Isolation of individual compounds from reverse phased fractions

The five fractions obtained from open phase column chromatography were subjected to HPLC analysis. The solvent phases were optimized based on the gradient HPLC results. The

fraction 1, fraction 2, fraction 3, fraction 4 and fraction 5 were run in 1% methanol, 5% methanol, 10% acetonitrile, 30% methanol and 20% acetonitrile in (0.5% HCOOH) acidified water respectively. The column used was YMC-TRIART $C_{18}$-(4.6ID X 250mm), at a flow rate of 1mL/min. The column temperature was maintained at 35°C and absorbance was recorded at 254nm.

### 2.11 NMR analysis

$^1$H, $^{13}$C NMR, and 2D NMR were recorded in JEOL ECA 500 spectrometer (500 MHz, JEOL Ltd, Tokyo Japan) and BRUKER Biospin AVANCEIII 600 (600 MHz, Bruker, Massachusetts, USA). The spectra were recorded in Acetone-d$_6$, DMSO-d$_6$, CD3OD, and D$_2$O at 298K. The expression of coupling constants is in hertz and chemical shifts are represented by δ in (parts per million) ppm scale. Tetramethylsilane (TMS) was added to the samples which served as an internal standard.

### 2.12 Mass spectra analysis

Analysis was performed on a UPLC system coupled to a QTOF-MS (Waters Xevo G2 QTof, Waters, Milford, MA, USA) instrument operated in electrospray ionization (ESI) mode at a mass resolution of 20,000 and controlled by MassLynx 4.1 software. An Acquity UPLC BEH C18 column (2.1 mm I.D. x 100 mm, 1.7 μm, Waters, USA) at 35°C was used for chromatographic separation. The sample (1 μL) was injected using an autosampler. The mass spectrometer was calibrated with 0.5 mM sodium formate. Leucine enkephalin (2 μg/mL, m/z 554.2615 in negative mode) was used as lock spray at a flow rate of 10 μL/min. The collision energy equaled 6V. The source parameters were as follows: capillary voltage 2.5kV, sampling cone voltage 30V, extraction cone voltage 4V, source temperature 150°C, desolvation temperature 500°Cgas flow 1000L/h, cone gas flow 50L/h. The mass spectrum analysis was done using Waters Xevo G2 QTof. The column used is AQUITY UPLC BEH C18 1mm ID x 100mm.

### 2.13 Spectroscopic and mass spectrum data of compounds 1–13

**Compound 1: -$C_7H_{10}O_5$**: Negative ESI-MS $m/z$ = 173.0025 [M-H]$^-$, $^1$H NMR (600MHz, D$_2$O): δ$_H$6.82 (1H, brt, $J$ = 2.4Hz), 4.45 (1H, brt, $J$ = 4.2Hz), 4.04 (1H, m), 3.78 (1H, q, $J$ = 8.4, 4.2Hz), 2.75 (1H, overlap), 2.23 (1H, dd, $J$ = 18, 6.6Hz)

**Compound 2: -$C_{14}H_{12}O_{11}$**: Negative ESI-MS $m/z$ = 336.9633 [M-H]$^-$, 337[M-H$_2$O-H]$^-$, $^1$H NMR (600MHz, D$_2$O): δ$_H$7.2 (1H,s), 5.4 (1H, d, $J$ = 1.2Hz), 4.01 (1H, dd,$J$ = 7.8, 1.2Hz), 3.3 (1H, ddd, $J$ = 14.4, 8.4, 6Hz), 2.9 (1H, dd, $J$ = 17.4, 8.4Hz), 2.6 (1H, dd, $J$ = 17.4, 6Hz).

**Compound 3: -$C_7H_6O_5$**: Negative ESI-MS $m/z$ = 168.9744 [M-H]$^-$, $^1$H NMR (600MHz, D$_2$O): δ$_H$7.08 (2H, s, H-2, H-6).

**Compound 4: -$C_6H_6O_3$**: Negative ESI-MS $m/z$[M+H]$^+$ = 108.9870, $^1$H NMR (600MHz, D$_2$O): δ$_H$9.4 (1H, s), 7.5 (1H, brd, $J$ = 3.35Hz), 6.6 (1H, brd, $J$ = 3.4Hz), 4.8 (1H, s), 4.6 (1H, s).

**Compound 5: -$C_7H_6O_4$**: Negative ESI-MS $m/z$ [M-H]$^-$ = 153.0059, $^1$H NMR (600MHz, DMSO-d$_6$): δ$_H$7.3 (1H, brd, $J$ = 1.8Hz), 7.2 (1H, dd, $J$ = 8.4, 1.8Hz), 6.7 (1H, d, $J$ = 8.4Hz).

**Compound 6: -$C_{14}H_{14}O_9$**: Negative ESI-MS $m/z$ [M-H]$^-$ = 325.0143, $^1$H NMR (600MHz, Acetone-d$_6$): δ$_H$7.08 (2H, s), 6.8 (1H, s), 5.35 (1H, dd, $J$ = 11.4, 4.2Hz), 4.49 (1H, s), 4.02 (1H, m), 2.8 (1H, overlap), 2.4 (1H, dd, $J$ = 18.6, 4.8Hz).

$^1$H NMR (600MHz, Acetone-d$_6$ + D$_2$O): δ$_H$7.1 (2H, s), δ6.8 (1H, s), 5.3 (1H, m), 4.5 (1H, brs), 4.04 (1H, overlap), 2.86 (1H, dd, $J$ = 17.4, 3.6Hz), 2.4 (1H, dd, $J$ = 19.2, 4.8Hz).

**Compound 7: -$C_{14}H_{14}O_9$**: Negative ESI-MS $m/z$ [M-H]$^-$ = 325.0257, $^1$H NMR (600MHz, Acetone-d$_6$): δ$_H$7.1 (2H, s), 6.8 (1H, brd, J = 3Hz), 5.78 (1H, m), 4.19 (1H, dd, $J$ = 12.6, 4.8Hz), 4.03 (1H, dd, $J$ = 6.6, 4.2Hz), 2.8 (1H, overlap), 2.32 (1H, m).

**Compound 8: -$C_8H_8O_5$**: Negative ESI-MS $m/z$ [M-H]$^-$ = 183.2640, $^1$H NMR (600MHz, CD$_3$OD): δ$_H$7.055 (2H, s, H-2, H-6), 3.82 (3H, s).

**Compound 9: -$C_{27}H_{22}O_{18}$**: Negative ESI-MS $m/z$ [M-H]$^-$ = 634.0134, [2M-H]$^-$ = 1267.0637, $^1$H NMR (600MHz, CD$_3$OD): δ$_H$7.055 (2H, s), 6.69 (1H, s), 6.66 (1H, s), 6.36 (1H, brs), 4.9 (1H, t, $J$ = 10.8Hz), 4.8 (1H, overlap), 4.5 (1H, t, $J$ = 9Hz), 4.4 (1H, brt, $J$ = 1.4Hz), 4.15 (1H, dd, $J$ = 10.8, 7.8Hz), 3.98 (1H, brd, $J$ = 1.2Hz).

**Compound 10: -$C_{27}H_{24}O_{18}$**: Negative ESI-MS $m/z$ [M-H]$^-$ = 634.9866, [2M-H]$^-$ = 1271.1479, $^1$H NMR (600MHz, CD$_3$OD): δ$_H$7.16, 7.14, 7.1 (each 2H, s), 5.8 (1H, d, $J$ = 8.4Hz), 5.2 (1H, t, $J$ = 9Hz), 4.59 (1H, dd, $J$ = 11.4, 1.2Hz), 4.4 (1H, dd, $J$ = 12.6, 6Hz), 3.89 (1H, ddd, $J$ = 9.6, 4.8, 2.4Hz), 3.7–3.8 (2H, m).

**Compound 11: -$C_{41}H_{30}O_{27}$**: Negative ESI-MS $m/z$ [M-H]$^-$ = 953.5759, [2M-H]$^-$ = 1907.0813, $^1$H NMR (600MHz, CD$_3$OD): δ$_H$7.47 (1H, s), 7.07 (2H, s), 6.84 (1H, s), 6.63 (1H, s), 6.5 (1H, s), 5.8 (1H, brs), 5.39 (1H, s), 5.23 (1H, s), 5.04 (1H, dd, $J$ = 6.6, 1.2Hz), 4.3 (1H, dd, $J$ = 10.8, 7.8Hz), 3.8 (1H, ddd, $J$ = 11.89, 4.9, 3.6, 1.5Hz), 3.72 (1H, brs), 2.2 (1H, dd, $J$ = 17.4, 3.6Hz), 2.1 (1H, dd, $J$ = 16.8, 11.4Hz).

**Compound 12: -$C_{41}H_{32}O_{27}$**: Negative ESI-MS $m/z$ [M-H]$^-$ = 954.9651, [2M-H]$^-$ = 1911.1218, $^1$H NMR (600MHz, CD$_3$OD): δ$_H$7.517 (1H, s), 7.18, 7.11, 6.96 (2H each, s, galloyl group), 6.5 (1H, d, $J$ = 1.8Hz), 6.24 (1H, brs), 5.4 (1H, t, $J$ = 5.4Hz), 5.11 (1H, dd, $J$ = 7.2Hz, 1.8Hz), 5.05 (1H, d, $J$ = 3.6Hz), 4.8 (1H, dd, $J$ = 5.4, 2.4Hz), 4.7 (1H, t, J = 6.6Hz), 4.6 (1H, dd, $J$ = 11.4, 7.2Hz), 3.9 (1H, dd, $J$ = 10.2, 4.2Hz), 2.2 (2H, d, $J$ = 12Hz).

**Compound 13: -$C_{14}H_6O_8$**: Negative ESI-MS $m/z$ [M-H]$^-$ = 300.9498, $^1$H NMR (600 MHz, Acetone-d$_6$): δ$_H$7.59 (2H, s).

## 2.14 Identification of the most active ingredient in HCAE

The concentrations of each compound identified from HCAE were evaluated by calculating area under the peak from gradient HPLC chromatogram. Based on the composition of HCAE obtained, different mixtures such as CM, C1′-C13′ were made. CM (Complete mixture) contained all the identified compounds which was formulated as per the composition. Mixture C1′ was made using all identified compounds except compound C1. Mixture C2′ was made using all identified compounds except compound C2 and so forth. The IC$_{50}$ values of all the resultant mixtures were evaluated using MTT assay to find out the variations in their ability to kill cancer cells in a dose dependent manner.

## 2.15 Cell cycle analysis of cancer cells treated with Haritaki churna aqueous extract

Approximately 5 x 10$^5$ cells were seeded in a 6 well plate one night before the day of the experiment. The cells were treated with HC aqueous extract at their respective IC$_{50}$ concentrations against HCT-116, DLD1, and HT-29. The cells were treated using serum-free media for 24 h, and cell cycle analysis was performed. FCS express software was used to analyse the different phases in the distribution of cells.

## 2.16 Live and apoptotic cell staining by acridine orange/propidium iodide method

The live and apoptotic cells were stained by acridine orange and propidium iodide as described previously [22, 23]. Approximately 1x10$^5$ cells were seeded in a 24-well plate one night before the day of the experiment. The cells were treated with HC aqueous extract in serum-free media at respective IC$_{50}$ concentrations for HCT-116, DLD1, and HT-29 for 48 h. The healthy, early apoptotic, late apoptotic, and dead cells were analysed by using quadrant analysis.

## 2.17 DNA fragmentation assay

Approximately $5 \times 10^5$ cells were seeded in 100 mm plates one night before the day of the experiment. The cells were treated with respective $IC_{50}$ concentrations for HCT-116, DLD1, and HT-29 cells in serum-free media for a period of 48 h. The cells grown in serum-free media were considered as control. DNA fragmentation analysis was performed as described previously [22, 23]. The DNA laddering was observed under UV light and images were taken in a BIO-RAD chemidoc[TM] imaging system.

## 2.18 Western blotting

DLD1 $(1*10^6)$ cells were seeded in a 6 well plate prior to the day of the experiment. Cells were washed twice with PBS and were treated with ellagic acid at concentrations of 2.5, 5, 10, 15 & 25 µg/ml for 24 and 48 hours. Cells were harvested and lysed with RIPA lysis buffer and the protein content was measured using standard protein assays. Proteins were separated on a 10% SDS-polyacrylamide gel and then transferred to nitrocellulose membrane (Bio-Rad cat. # 162–0112) on a Trans-Blot Turbo (Bio-Rad). The membrane was blocked with 5% BSA for 1–2 hours at room temperature. The blots were then incubated with primary antibodies (RIP (1:2000), p-RIP (1:1000), MLKL (1:2000), p-MLKL (1:1000), Cleaved caspase 3 (1:2000), Caspase 3 (1:2000), Cleaved caspase 8 (1:2000), Caspase 8 (1:2000), Bcl2 (1:2000), PARP-1 (1:2000), Cytochrome-C (1:1000), Cyclin D1 (1:5000), CyclinA2 (1:4000) GAPDH (1:5000) and β-actin (1:5000)) overnight at $4^\circ$C followed by incubation with appropriate HRP conjugated secondary antibodies for 1–2 hours at room temperature. Proteins bands were analysed by using Bio-Rad Clarity[TM] Western ECL substrate kit and images were developed in Bio-Rad chemiDoc system.

## 2.19 Statistical analysis

Sample values are expressed as mean ± standard deviation (SD), N = 3. All the $IC_{50}$ values were calculated using a non-linear curve fit model in ORIGIN 2019 software.

## 3 Results and discussion

### 3.1 Haritaki churna aqueous extract has anti-cancer activity

In Ayurveda, it is recommended to soak Haritaki Churna (HC) into cold water for overnight and then the water extract can be consumed orally for maximum therapeutic effect. Considering this aspect, we have prepared the aqueous extract of HC as described in "section **2.2**". The aqueous extract of HC was tested against different cancer cells to evaluate the anti-cancer potential of formulation as described. Cells were treated with different concentrations of aqueous extract of HC for 48 hours and cell viability was measured by MTT assay. Cancer cells treated with aqueous extract of HC exhibits dose dependent loss of cellular viability and alternations in cellular morphology. The aqueous extract is giving anti-cancer activity against MDAMB-23, HeLa, & MG63 cells with an $IC_{50}$ of 53.1 ± 4.96 µg/ml, 79.35 ± 4.95 µg/ml, & 97.04 ± 4.09 µg/ml respectively (**Table 1**).

### 3.2 Haritaki churna aqueous extract induces dose-dependent loss of cellular viability in colon cancer cells

The Haritaki churna in the ayurvedic literature is well known for treating GI-track-related disorders. The concentration of bioactive agents present in HC will be very high in GI track and we have asked if the ayurvedic formulation could have potential to exhibit anti-cancer activity

**Table 1. IC$_{50}$ values of Haritaki churna aqueous extract for different cell lines.**

| Cell line | Type | IC$_{50}$ (µg/ml)±S.D |
|---|---|---|
| MDAMB-231 | Breast cancer | 53.1 ± 4.96 |
| HeLa | Cervical cancer | 79.35 ± 4.95 |
| MG-63 | Osteosarcoma | 97.04 ± 4.09 |
| HCT-116 | Colorectal | 92.69 ± 7.07 |
| DLD1 | Colorectal | 70.41 ± 6.35 |
| HT-29 | Colorectal | 379.93 ± 5.29 |
| HEK-293 | Embryonic | >300 |
| PBMCs | Blood mononuclear | >500 |

Note: Cancer cells, HEK-293 and PBMC (Peripheral blood mononuclear cells extracted from fresh blood samples) were treated with HCAE (0–1000 µg/ml) for a period of 48 h. The cellular viability was measured by MTT assay as described. The IC$_{50}$ values were calculated using non-linear curve fit model in Origin 9 PRO software.

against colon cancer. To explore such question, we have treated the colon cancer cells HCT-116, DLD-1 and HT29 with aqueous extract of HC and measured the cellular viability by MTT assay. Cancer cells were treated with different concentration of HC and it exhibits dose-dependent killing of cancer cells (**Fig 1**). Among the three colon cancer cell lines, HCAE showed maximum activity towards DLD1 with an IC$_{50}$ of 70.41 ± 6.35 µg/ml, whereas it showed IC$_{50}$ of 92.69 ± 7.07 µg/ml against HCT-116and 379.93 ± 5.29 µg/ml against HT-29 (**Table 1**). In addition, treated cells were exhibiting membrane blebbing and distorted morphology indicating extreme stress inside the cells (**Fig 1**). The data in **Fig 1** and **Table 1** clearly highlight that the HC has bioactive agents with anti-cancer activity and it can be useful to treat colorectal cancer in patients.

## 3.3 Haritaki churna aqueous extract is safe for normal human cells

The major drawback of studies with ayurvedic formulation is systematic exploration of many pharmacological properties such bio-availability, distribution safety and toxicity etc. Before exploring the composition and purification of bioactive anti-cancer agents present in HC aqueous extract, we have performed safety and toxicity analysis. We have used two different models to test the safety and toxicity of HC aqueous extract. In model 1, we have tested the safety of the HC extract using primary cells. The use of peripheral blood mononuclear cells (PBMCs) has been widely accepted as an in-vitro model for testing the safety of anti-cancer drugs [31–34]. PBMCs were isolated from human blood as described and sub-cultured in DMEM complete medium. PBMCs were treated with different concentration of HC aqueous extract (0–500µg/ml) and cellular viability was measured by MTT assay. We found that HCAE is not affecting cellular viability of PBMCs even at a very high concentration of 500µg/ml (**Fig 2A**). Alteration in cellular morphology is a primary event to monitor the safety of the anti-cancer drug. Observation of HC aqueous extract treated PBMCs indicate no significant alternation in cellular morphology (**Fig 2B**). In model 2, non-cancerous cell line HEK293 was used to test the toxicity of HC aqueous extract. HEK 293 cells were treated with different concentration of HC extract and cellular viability was measured by MTT assay. HC aqueous extract had not affected the cellular viability of HEK-293 cells up to concentration of 300 µg/ml. Blebbing in HEK-293 cells treated with higher concentration (500µg/ml) of HCAE was observed (**Fig 2B**). However, these protrusions have been found in HEK-293 cells at doses greater than those necessary to kill colorectal cancer cells. Our results comply with the study done by suganthy

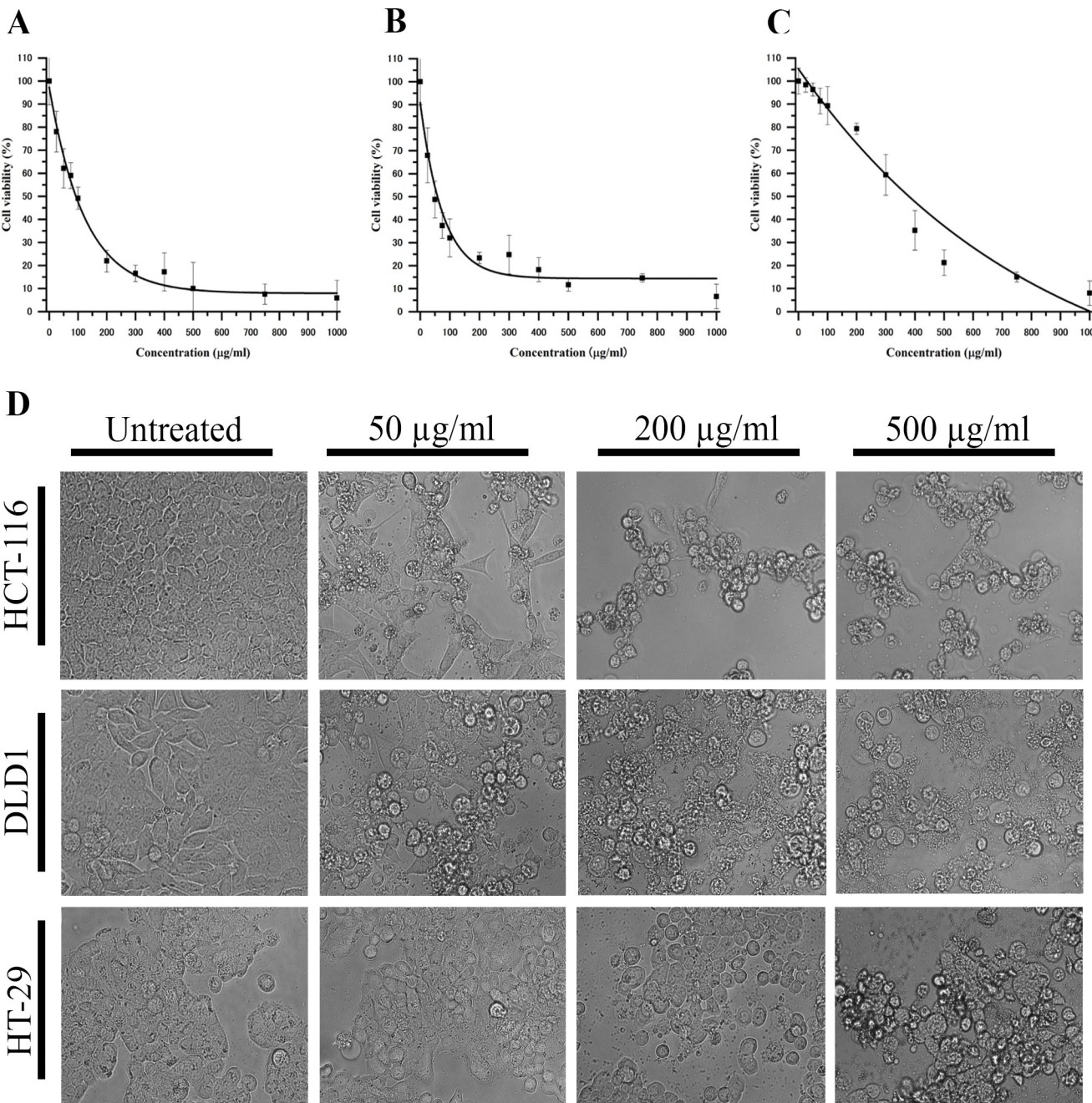

**Fig 1. Haritaki churna aqueous extract is effective against colon cancer cells.** Colon Cancer cells were treated with different concentration of Haritaki churna aqueous extract (0–1000 µg/ml) and the cell viability was measured by MTT assay. The MTT curve presented in **(A)**-**(C)** refer to the treatment of HCAE to HCT-116, DLD1 and HT-29 cells respectively. **(D)** Cells were observed under microscope and different images were acquired from different fields using Cytell cell imaging system (GE Healthcare). Representative images from each treatment were presented.

et al, where the activity of methanolic extracts of *Terminalia chebula* fruit (TCF), *Terminalia arjuna* bark (TAB), and 7-methylgallate(7-MG) was checked on PBMCs [35]. They showed that there was no significant decrease in the viability and membrane integrity of PBMCs upon treatment by TCF, TAB & 7MG even at concentrations of 500–2000 µg/ml.

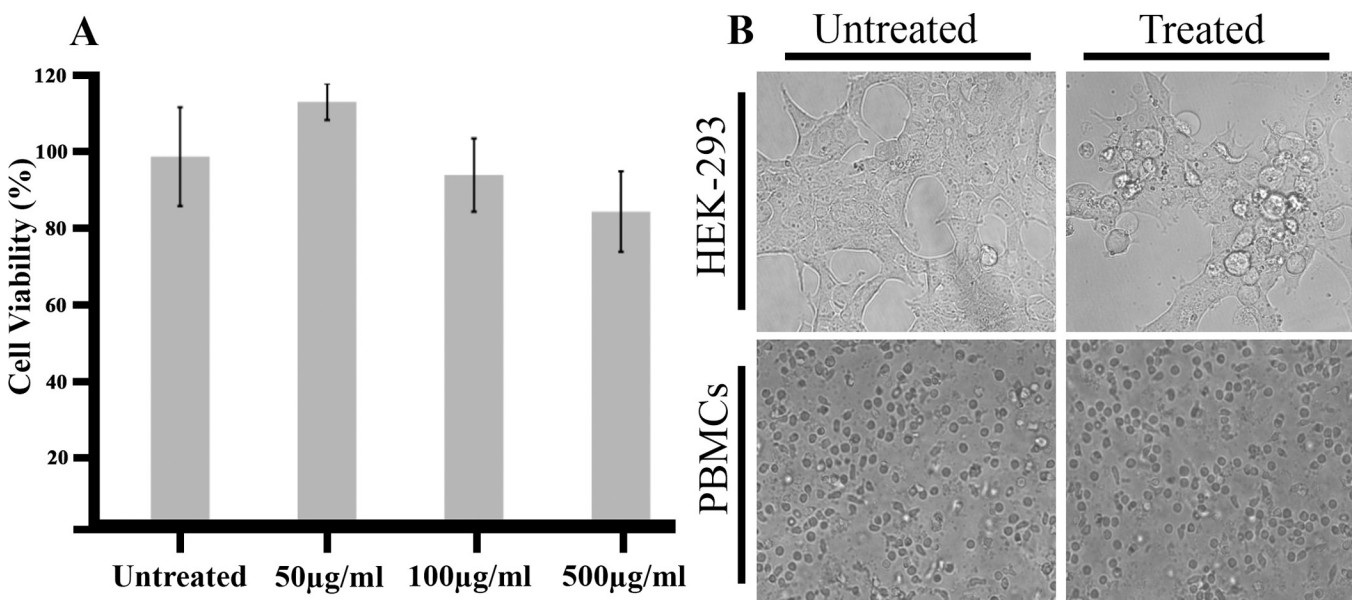

**Fig 2. Haritaki churna aqueous extract is safe for human consumption. (A)** The peripheral blood mononuclear cells were isolated from healthy volunteer as described. The cells treated with Haritaki churna aqueous extract for a period of 48 hours and their cell viability was measured by MTT assay as given in the procedure. **(B)** HEK-293 and PBMC Cells were observed and 5 different random images were taken using Cytell cell imaging system (GE Healthcare). Representative images are shown for untreated and treated cells.

### 3.4 Aqueous extract of Haritaki churna has several bioactive agents

One gram of Haritaki churna yielded 539.6 ± 14.5 mg of aqueous extract when extracted as per the procedure mentioned in section **2.2**. The hydrophilic contents present in the ayurvedic formulation are the principal ingredients as the water extract constitutes approximately more than 50% of crude Haritaki churna powder. The qualitative phytochemical composition of the aqueous extract of HCAE was determined using several biochemical assays. The findings indicated that polyphenols, protein, alkaloids, flavonoids, and terpenoids are present. The polyphenol group of compounds were present in the highest amount (**Fig 3**). The TPC of HCAE was found to be 395.9 mg gallic acid equivalents/gram dry weight of extract. The gradient HPLC chromatogram of aqueous extract of Haritaki churna also confirmed the presence of polyphenols when it was spiked at 254nm. The HPLC chromatogram has 13 prominent peaks and 11 minor peaks. The major peaks have been labelled in the chromatogram (**Fig 3**). The several bioactive agents present in these peaks could be responsible for anti-cancer activity of the HC aqueous extract.

### 3.5 Fractionation and isolation of compounds from the aqueous extract of Haritaki churna

Further, the HC aqueous extract was serially eluted with solvents with varying polarity in an open column chromatography using HP20SS resin as described in section **2.8**. The yields of fractions **1–5** have was calculated to be 140.4 ± 15.6 mg, 47.5 ± 3.09 mg, 157.6 ± 16.6 mg, 81.2 ± 7.02 mg, 10.2 ± 2.6 mg (**S1 Data**). The schematic of the procedure for separating the fractions using open column chromatography is illustrated in **Fig 4A**. The chromatograms of different fractions were developed and the obtained fractions were subjected to isocratic HPLC analysis for setting up the condition for isolation of individual compounds from the fractions. The fraction **1** has the most hydrophilic compounds whereas fraction **5** contains the

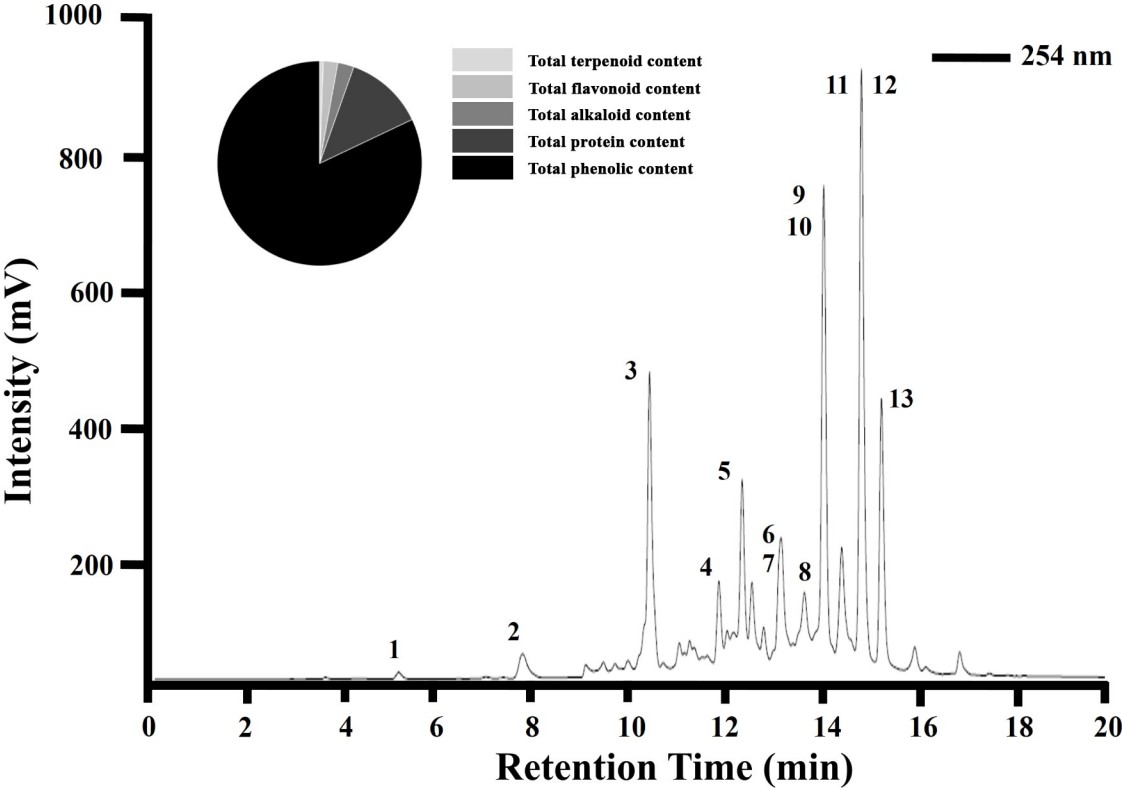

**Fig 3. Haritaki churna are composed of different types of bioactive compounds.** The gradient HPLC chromatogram shows the presence of different bioactive compounds in Haritaki churna aqueous extract at 254 nm.

most hydrophobic ones providing us a gradient advantage in separating compounds. The fraction **1** and fraction **4** contained two compounds each (compound **1** & **2** in fraction **1** and compound **11** & **12** in fraction **4**) whereas fraction **2** & **5** composed of one compound each. A copious number of compounds was present in fraction **3** which may be because of intermediary solubility in organic and aqueous solvents (**Fig 4**).

### 3.6 Structural elucidation of chemical compounds isolated from Haritaki churna aqueous extract (HCAE)

Chemical compounds present in HC aqueous extract was purified to homogeneity and purity was confirmed by spiking the compound in HPLC (**S2 Data**). The structure of purified compound was determined by spectroscopic techniques [1]H NMR (600 MHz) and [13]C NMR (600 MHz) along with 2D NMR techniques for some compounds. The molecular weight of compound was determined by using mass spectra UPLC system coupled to a QTOF-MS. Compound **1** showed a molecular ion peak at *m/z* 173.0025 [M-H][-]. The molecular formula $C_7H_{10}O_5$ was deduced using [1]H NMR, LC-MS analysis and was identified as shikimic acid (**Fig 5.1**) by comparing it with already published literature [36]. Compound **2** showed two molecular ions at *m/z* 336.9633 [M-H]—and 354.9868 [M-H][-]. The peak at *m/z* 336.9633 [M-H][-] is due to the loss of water molecules from the parent molecule because of high ionization energy. The [1]H, LC-MS analyses were used in deducing $C_{14}H_{12}O_{11}$ and are identified as chebulic acid (**Fig 5.2**) by comparing it with already published literature [37]. Compound **3** showed the highest peak at *m/z* 168.9744 [M-H][-]. $C_7H_6O_5$ was deduced as the molecular

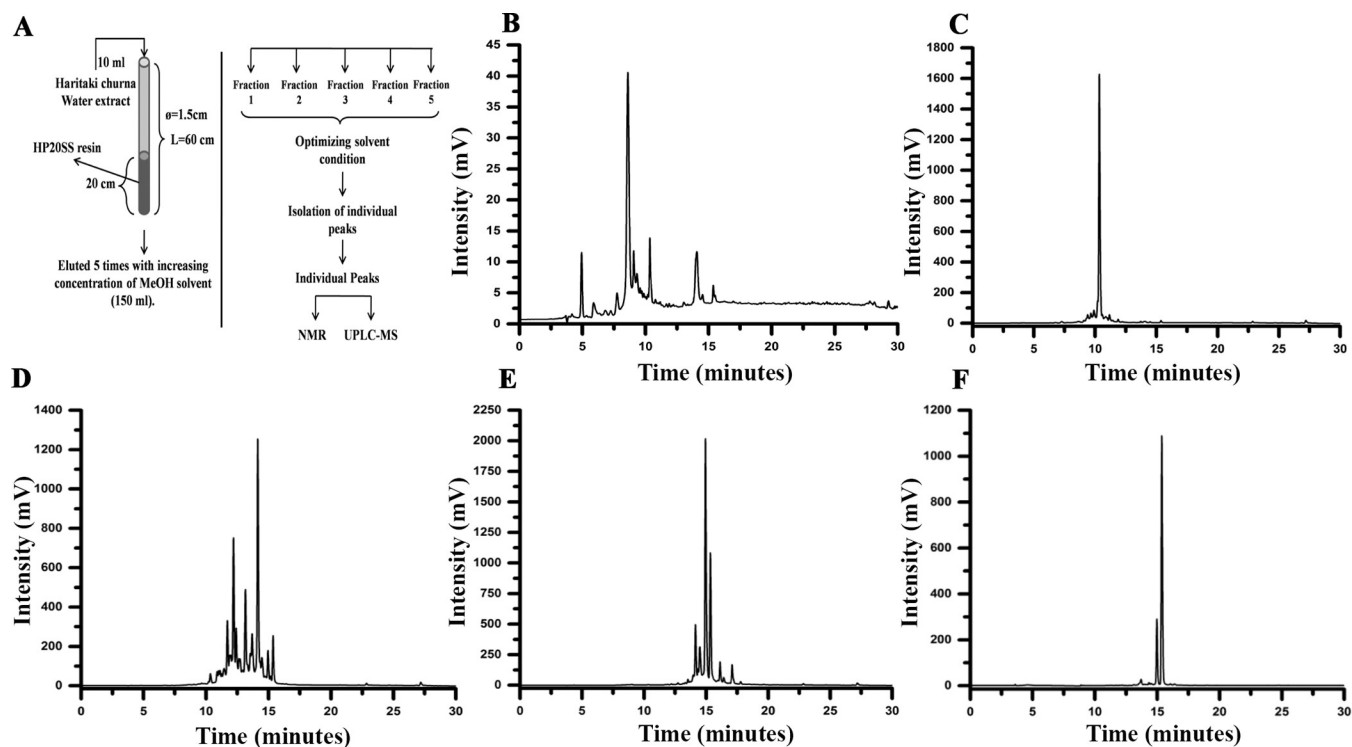

**Fig 4. Extraction and fractionation of Haritaki churna aqueous extract. (A)** Schematic of the procedure followed for reverse phase open column chromatography. **(B)** Chromatogram for fraction **1**. **(C)** Fraction**2 (D)** Fraction**3**. **(E)** Fraction**4 (F)** Fraction **5**.

formula by analysing the $^1$H NMR and was identified as gallic acid (**Fig 5.3**) by comparing it with already published literature [38]. Compound **4** showed molecular ion peaks at *m/z* 108.9870 [M+H]$^+$, 127.0182 [M+H]$^+$in positive mode. The peak at *m/z* 108.9870 [M]$^+$ could be because of the loss of water molecules from the parent molecule because of high ionization energy. Molecular formula $C_6H_6O_3$ was obtained by analysing $^1$H NMR and was identified as 5-hydroxymethylfurfural (**Fig 5.4**) by comparing it with already published literature [39]. Compound **5** showed the highest molecular ion peak at *m/z* 153.0059 [M-H]$^-$. The $^1$H and LC-MS analysis helped us deduce the molecular formula as $C_7H_6O_4$ and were identified as protocatechuic acid (**Fig 5.5**) after comparing it with published literature [40]. Major ion peak of compound **6** under negative mode showed a peak at *m/z* 325.0143 [M-H]$^-$. This ion was also accompanied by cluster ion at *m/z* 651.0958 [2M-H]$^-$. The fragment ions *m/z* 169.0021 [M-H]$^-$ and *m/z* 173.0442 [M-H]$^-$ corresponded to the gallic acid moiety and shikimic acid moiety, respectively. ESI-MS data of **7** was similar to **6**, the molecular weight was the same. Therefore, **6** and **7** were assumed positioning isomer each other. Based on 1H NMR analysis and comparing it to the previously published literature, the compound **6** and compound **7** were identified as 4-*O*-galloyl shikimic acid and 5-*O*-galloyl shikimic acid [41] (**Fig 5.6 & 5.7**). Compound **8** showed a molecular ion peak at *m/z* 183.2640 [M-H]$^-$. The molecular formula was deduced as $C_8H_8O_5$ with the help of $^1$H NMR analysis and was identified as methyl gallate (**Fig 5.8**) by comparing with published literature [42]. Compound **9** showed two molecular ion peaks at *m/z* 634.0134 [M-H]$^-$ and 1267.0637 [2M-H]$^-$. The $^1$H NMR analysis helped us in deducing the formula as $C_{27}H_{22}O_{18}$ and was identified as corilagin (**Fig 5.9**) by comparing it with published literature [43]. Compound **10** showed two molecular ion peaks at *m/z* 634.9866 [M-H]$^-$and 1271.1479 [2M-H]$^-$. After $^1$H NMR analysis, the molecular formula was deduced

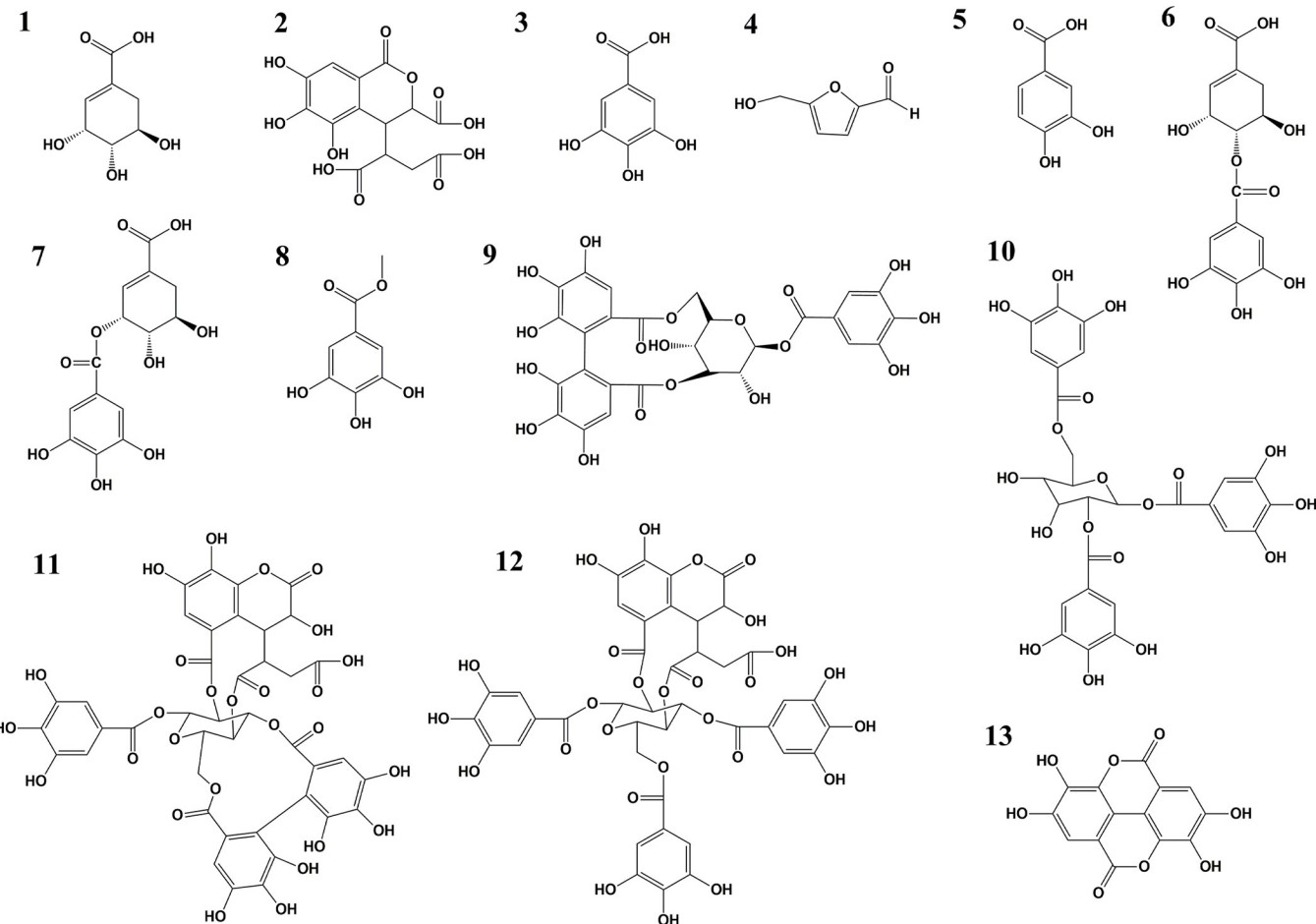

**Fig 5. Chemical structures of isolated compounds from aqueous extracts of Haritaki churna.** (**1**: Shikimic acid, **2**: Chebulic acid, **3**: Gallic acid, **4**: 5 hydroxymethyfurfural, **5**: Protocatechuic acid, **6**: 4-O-galloyl shikimic acid, **7**: 5-O-galloyl shikimic acid, **8**: Methyl gallate, **9**: Corilagin, **10**: 1,2,6-Tri-*O*-galloyl-β-D glucose, **11**: Chebulagic acid, **12**: Chebulinic acid, **13**: Ellagic acid).

as $C_{27}H_{24}O_{18}$ and was identified as 1, 2, 6, tri-*O*-galloyl β-D-glucose (**Fig 5.10**)after comparing it with published literature [43]. Compound **11** showed two molecular ion peaks at *m/z* 953.5759 [M-H]⁻ and 1907.0813 [2M-H]⁻. The $^1$H NMR analysis helped us in deducing the formula as $C_{41}H_{30}O_{27}$ and was identified as chebulagic acid (**Fig 5.11**) by comparing it with published literature [42]. Compound **12** showed two molecular ion peaks at *m/z* 954.9651 [M-H]⁻ and 1911.1218 [2M-H]⁻. The $^1$H NMR analysis helped us in deducing the formula as $C_{41}H_{32}O_{27}$ and was identified as chebulinic acid (**Fig 5.12**) by comparing it with published literature [42]. Compound **13** showed a molecular ion peak at *m/z* 300.9498 [M-H]⁻. The $^1$H NMR, LC-MS analysis and comparison of Ellagic acid standard on isocratic HPLC helped us identify compound **13** as ellagic acid (**Fig 5.13**). The NMR spectra and mass spectra of all compounds is shown in **S1 Data**. The composition, retention time, molecular formulae and the purity of the compounds isolated from HC aqueous extract is shown in **Table 2**.

## 3.7 Haritaki churna is stable in physiological gastric condition

The stability of bioactive natural compounds in biological fluid is very crucial for their activity and delivery to the action site. Haritaki Churna is the primarily an ayurvedic formulation to

treat diseases of GI track. The use of dissolution media such as SGF has helped researchers in the approval of effective and stable generic drugs which has been well documented [44]. The Haritaki churna was incubated in water and stimulated gastric fluid (SGF) and then we assess the stability of bioactive compounds and their anti-cancer activity. The HPLC chromatograms show that all major peaks were present in both chromatograms no considerable changes in the overall composition of the formulation even after 6 hours of incubation (**Fig 6A–6D**). In addition, we extracted the Haritaki churna in SGF to see if differential extraction could result in different proportions of compounds being extracted or whether it alters the yield of extraction. The yield was anyhow similar when extracted in water or SGF. The yield in water was 539.6 ± 14.5 mg whereas it was 541.2 ± 17.3 mg as per the extraction procedure mention in section 2.2 & 2.6. However, SGF extract chromatogram shows a decrease in the intensity of compounds 11, 12 & 13 and increase in the intensities of compounds 5, 9 & 10 but there is no change in composition of overall compounds present in both extracts (**Fig 6E & 6F**). The $IC_{50}$s of HCAE incubated in SGF with different time intervals are fundamentally close to the original HCAE $IC_{50}$ value (**Fig 6G**). Further, dissolution media such as FaSSIF and FeSSIF, containing bile salts, pancreatin at relevant pH conditions have been recognised and utilised to mimic the conditions in the stomach before and after meals [25]. In contrast to HCAE as the control, there were no significant differences in the chromatograms of the formulation incubated with FaSSIF and FeSSIF, indicating that HC was stable in both the fasting and fed phases (**Fig 7A–7C**). Additionally, the stability of the compounds present in HCAE was tested by incubating it in FaSSIF and FeSSIF at different time intervals. An increase in the intensity of compound 13 (ellagic acid) and a decrease in the intensity of the compounds 11 & 12 (Chebulagic acid and chebulinic acid) were observed indicating a degradation of those compounds in a time dependent manner (**Fig 7D & 7E**). It is well established that bulky polyphenols are known to be unstable at pH in the range of 5–7 [45]. The $IC_{50}$ values of HCAE incubated in FaSSIF and FeSSIF at various time intervals decreased (indicating an increase in anti-cancer activity) because a rise in compound 13's intensity (ellagic acid) was seen in HCAE with their incubations at different time intervals (**Fig 7F & 7G**). Given its stability in SGF at low pH circumstances and an increase in anti-cancer activity of HCAE in simulated fasting and fed state intestinal settings, it is reasonable to assume that Haritaki churna formulation can be employed as an anti-cancer drug due to its high bioavailability.

**Table 2. The phytochemicals isolated from Haritaki churna aqueous extract.**

| Peak No. | Phytochemicals | Molecular formula | Composition (%) | Retention time (min) | Molecular weight (Da) | Peak purity (%) |
|---|---|---|---|---|---|---|
| 1 | Shikimic acid | $C_7H_{10}O_5$ | 0.351 | 5.1 | 174 | 83.2 |
| 2 | Chebulic acid | $C_{14}H_{12}O_{11}$ | 2.168 | 7.99 | 336 | 91.2 |
| 3 | Gallic acid | $C_7H_6O_5$ | 19.781 | 10.37 | 170 | 98.9 |
| 4 | 5-Hydroxymethylfurfural | $C_6H_6O_3$ | 0.921 | 11.01 | 126 | 96.2 |
| 5 | Protocatechuic acid | $C_7H_6O_4$ | 2.839 | 11.89 | 154 | 98.5 |
| 6 | 4-O-galloyl shikimic acid | $C_{14}H_{14}O_9$ | 3.247 | 13.12 | 326 | 98.2 |
| 7 | 5-O-galloyl shikimic acid | $C_{14}H_{14}O_9$ | 3.247 | 13.12 | 326 | 97.1 |
| 8 | Methyl gallate | $C_8H_8O_5$ | 5.202 | 13.8 | 184 | 94.5 |
| 9 | Corilagin | $C_{27}H_{22}O_{18}$ | 15.919 | 14.08 | 634 | 92.2 |
| 10 | 1,2,6 Tri-O-galloyl-β-D-glucose | $C_{27}H_{24}O_{18}$ | 6.861 | 14.35 | 636 | 82.1 |
| 11 | Chebulagic acid | $C_{41}H_{30}O_{27}$ | 11.838 | 15.05 | 954 | 97.5 |
| 12 | Chebulinic acid | $C_{41}H_{32}O_{27}$ | 7.889 | 15.05 | 956 | 92.6 |
| 13 | Ellagic acid | $C_{14}H_6O_8$ | 12.662 | 15.5 | 302 | 98.2 |
| 14 | Rest of the compounds | --------- | 7.08 | --------- | --------- | --------- |

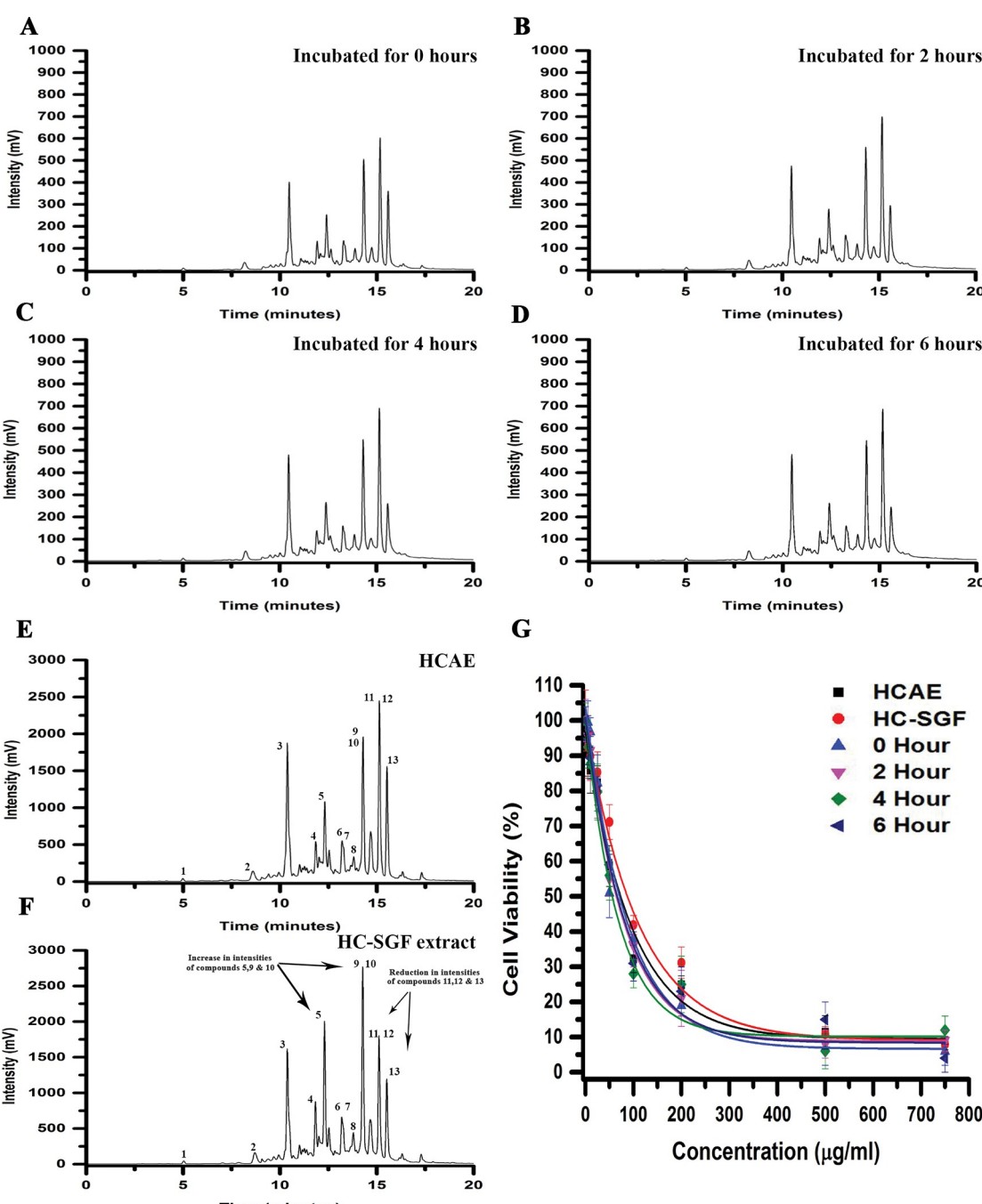

**Fig 6. Haritaki churna is stable under simulated gastric environment conditions.** 100 mg of lyophilised HCAE was dissolved in 1 ml of SGF and incubated for different time intervals at 37°C. (**A**) Gradient HPLC chromatogram of HCAE incubated in SGF for 0 Hours. (**B**) Gradient HPLC chromatogram of HCAE incubated in SGF for 2 Hours. (**C**) Gradient HPLC chromatogram of HCAE incubated in SGF for 4 Hours. (**D**) Gradient HPLC chromatogram of HCAE incubated in SGF for 6 Hours. The Haritaki churna extracted in water and SGF were prepared as mentioned before. (**E**) Gradient HPLC chromatogram of HC extracted in water. (**F**) Gradient HPLC chromatogram of HC extracted in SGF. (**G**) HCAE extract retains the biological activity in SGF. DLD1 cells were treated with different concentration of HCAE and cellular viability was measured. MTT cell viability results seem to be coherent with the chromatogram data. The $IC_{50}$s of HCAE incubated with 0 Hour, 2 Hour, 4 Hour, and 6 Hour are 67.08± 5.2 μg/ml, 63.9 ± 6.3 μg/ml, 65.76 ± 4.66 μg/ml, and 63.4 ± 5.1 μg/ml respectively.

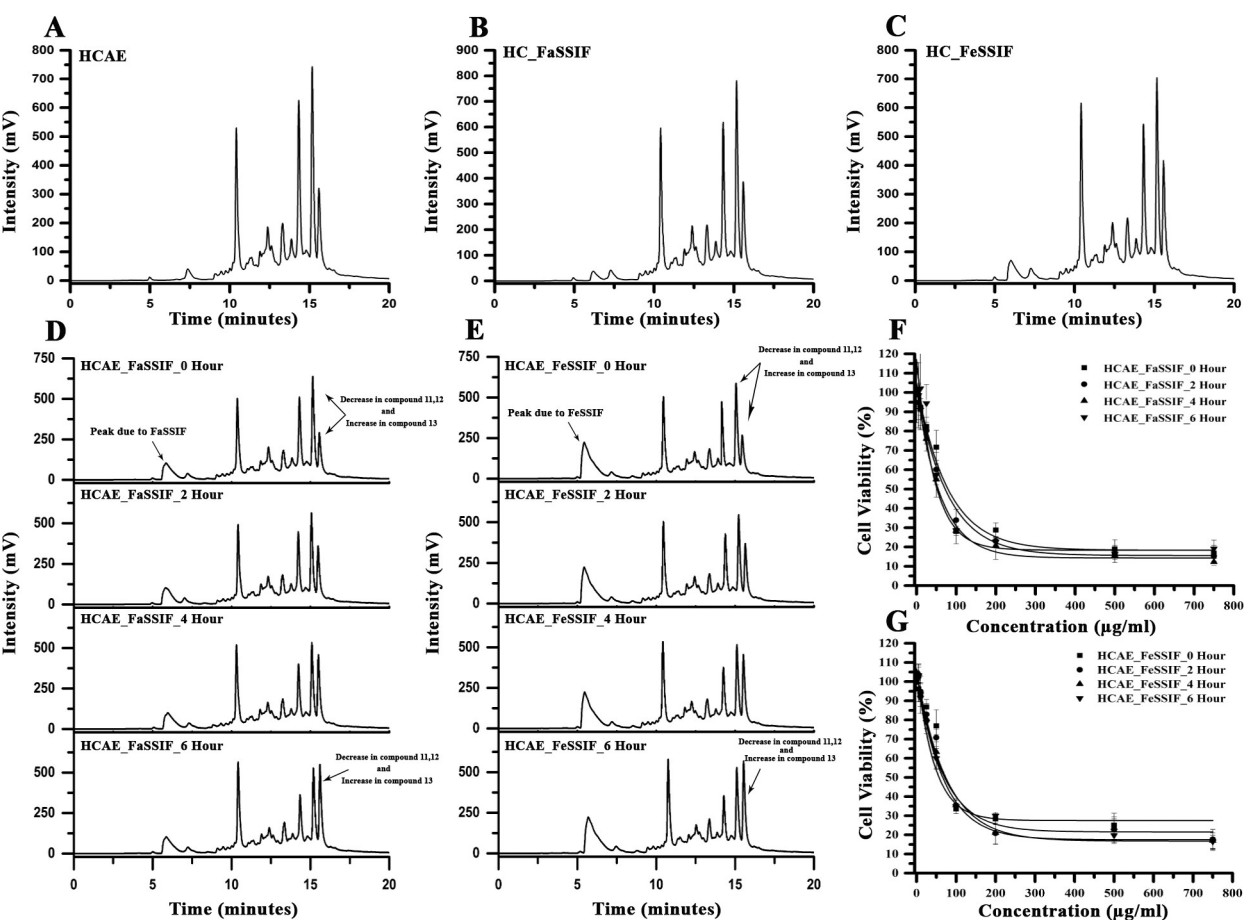

**Fig 7. Haritaki churna is stable under simulated intestinal environment conditions: 100mg of Haritaki churna was dissolved in water, FaSSIF and FeSSIF at 37°C. (A)** Gradient HPLC chromatogram of HC incubated in water. **(B)** Gradient HPLC chromatogram of HC incubated in FaSSIF. **(C)** Gradient HPLC chromatogram of HC incubated in FeSSIF. **(D)** Gradient HPLC chromatogram of HCAE incubated in FaSSIF for 0, 2, 4 & 6 Hours. **(E)** Gradient HPLC chromatogram of HCAE incubated in FeSSIF for 0, 2, 4 & 6 Hours. **(F)** DLD1 cells were at different concentrations with HCAE incubated in FaSSIF at different time intervals. The $IC_{50}$ values of HCAE incubated with FaSSIF for 0 Hour, 2 Hour, 4 Hour, and 6 Hour are 77.04 ± 6.4 μg/ml, 72.1 ± 5.4 μg/ml, 66.6 ± 3.7 μg/ml, and 59.6 ± 6.6 μg/ml respectively. **(G)** DLD1 cells were at different concentrations with HCAE incubated in FeSSIF at different time intervals. The $IC_{50}$ values of HCAE incubated with FeSSIF for 0 Hour, 2 Hour, 4 Hour, and 6 Hour are 70.8 ± 4.5 μg/ml, 74.1 ± 4.5 μg/ml, 66.4 ± 6.01 μg/ml, and 57.1 ± 3.7 μg/ml respectively.

### 3.8 Haritaki churna aqueous extract has bioactive compounds with anti-cancer activity

Haritaki churna crude extract has several bioactive compounds and now we have tested individual isolated to explore the role of these compounds in observed anti-cancer activity in crude extract (**Fig 1**). The HC aqueous extract was fractionated into five major fractions from reverse phase open column chromatography (**Table 3**). The cancer cells (HCT-116, DLD1, and HT-29) were treated with different number of fractions **1–5** and cellular viability was measured by MTT assay as described in "material and methods". There were 13 major compounds identified from the five different fractions. Shikimic acid and Chebulic acid identified from fraction **1** didn't affect the cellular viability of cancer cells. Gallic acid was the primary compound present in fraction **2**. Gallic acid extracted from fraction 2 showed anti-cancer activity with an $IC_{50}$ values in the range of 17–37 μg/ml against different cancer cell lines. Collective evidence has shown that Gallic acid is shown to have anti-cancer activities against a variety of cancer cell lines and our experimental results match with those of previously published results

**Table 3. Anti-cancer activity of different fractions and pure phytochemicals against colon cancer.**

| S No | Fraction/ Compound name | HCT-116 IC$_{50}$(µg/ml) | DLD1 IC$_{50}$(µg/ml) | HT-29 IC$_{50}$(µg/ml) |
|------|------------------------|--------------------------|----------------------|------------------------|
| 1 | HCAE | 92.69 ± 7.07 | 70.41 ± 6.35 | 379.93 ± 5.29 |
| 2 | HC-0 Fraction | > 500 | > 500 | > 500 |
| 3 | HC-20 Fraction | 33.71 ± 4.24 | 15.61 ± 4.89 | 43.57 ± 4.98 |
| 4 | HC-50 Fraction | 155.5 ± 7.59 | 79.39 ± 4.83 | 141.43 ± 4.66 |
| 5 | HC-80 Fraction | 71.29 ± 5.4 | 36.53 ± 3.77 | 115.7 ± 4.8 |
| 6 | HC-100 Fraction | 14.03 ± 3.46 | 20.14 ± 3.99 | 13.32 ± 3.48 |
| 7 | Shikimic acid | > 500 | > 500 | > 500 |
| 8 | Chebulic acid | > 300 | > 300 | > 300 |
| 9 | Gallic acid | 37.1 ± 4.56 | 18.1 ± 4.63 | 35.06 ± 5.31 |
| 10 | 5-Hydroxymethylfurfural | > 500 | > 500 | > 500 |
| 11 | Protocatechuic acid | > 300 | > 300 | > 300 |
| 12 | 4-O-galloyl shikimic acid | 311.2 ± 6.2 | 179.6 ± 4.6 | 161 ± 4.79 |
| 13 | 5-O-galloyl shikimic acid | 281.5 ± 5.2 | 136.8 ± 4.8 | 203.7 ± 5.8 |
| 14 | Methyl gallate | 38.68 ± 4.51 | 37.8 ± 3.39 | 30.71 ± 3.76 |
| 15 | Corilagin | 23.91 ± 3.82 | 63.08 ± 4.21 | 157.39 ± 4.34 |
| 16 | 1,2,6 Tri-O-galloyl β-D glucose | 100.11 ± 5.29 | 157.5 ± 7.64 | 114.22 ± 4.49 |
| 17 | Chebulagic acid | > 250 | > 250 | > 400 |
| 18 | Chebulinic acid | 54.04 ± 5.41 | 22.63 ± 4.21 | 122.4 ± 4.02 |
| 19 | Ellagic acid | 10.08± 3.46 | 17.39± 4.89 | 13.1± 4.81 |

**Foot note:** Shikimic acid, chebulic acid, gallic acid, 5-hydroxymethylfurfural, protocatechuic acid, 4-O-galloyl shikimic acid, 5-O-galloyl shikimic acid, methyl gallate, 1, 2, 6, Tri-O-galloyl β-D-glucose, and corilagin, were dissolved in water whereas chebulagic acid, chebulinic acid, and Ellagic acid were dissolved in DMSO.

[46–50]. The five major compounds present in fraction **3** are 5-Hydroxymethylfurfural, proto-catechuic acid, Methyl-gallate, Corilagin, and 1, 2, 6 Tri-O-galloyl-β-D-glucose. 5-hydroxy-methylfurfural and proto-catechuic acid didn't affect the cellular viability of cancer cells. Methyl-gallate which is similar in structure to gallic acid showed the anti-cancer activity with IC$_{50}$ in the range of 37–52 µg/ml against different cancer cell lines. Methyl-gallate is a relatively new compound compared with that of gallic acid, but there is still mounting evidence that suggests it exerts anti-cancer activity against a variety of cancer cell lines [51–55]. Apart from Gallic acid and Methyl-gallate, Corilagin also exhibited killing of cancer cells with an IC$_{50}$ in the range of 23–157µg/ml against different cancer cell lines. Corilagin has been tested against ovarian, nasopharyngeal, osteosarcoma, colon, hepatocellular, lung adenocarcinoma, etc. which have shown IC$_{50}$ in the range of 0.25–19 µg/ml [56–60]. 1, 2, 6 Tri-O-galloyl-β-D-glucose which is structurally similar to Corilagin also showed loss of cellular viability in cancer cells but at a higher IC$_{50}$ in the range of 100–175 µg/ml. Chebulagic acid and chebulinic acid were the two compounds isolated from fraction **4**. Chebulagic acid didn't affect the cellular viability of cancer cells whereas chebulinic acid was effective against all cancer cell lines having an IC$_{50}$ concentration in the range of 22–122 µg/ml against different cancer cells. Recent studies done on chebulinic acid extracted from *Terminalia chebula* Retz. showed that chebulinic acid exerts anti-cancer activity against HR8348, LoVo & LS174T (colorectal cancer cell lines) having IC$_{50}$ in the range of 38–40 mg/L and 53.2±0.16 µM for HOS-1 cells [61]. Fraction **5** contained Che-bulinic acid and Ellagic acid which showed IC$_{50}$ concentration in the range of 10–20µg/ml. Ellagic acid isolated from fraction **5** also showed IC$_{50}$ in the range of 10–20 µg/ml. The experimental IC$_{50}$ values of ellagic acid are in coherence with published literature [62, 63]. The IC$_{50}$ values for different fractions and isolated compounds have been listed in the **Table 3** and dose-dependent response graphs used to calculate the anti-cancer IC$_{50}$ are given in the **S3**

**Data**. Gallic acid, methyl gallate and ellagic acid were found to be the most active polyphenols from HCAE based on their anti-cancer activity. The individual compositional make-up of gallic acid, methyl gallate, and ellagic acid were measured using their standard curves and determined to be 227.1 ± 24.3 μg, 124.1 ± 9.4 μg, and 67 ± 6.3 μg respectively per 1mg of HCAE (**S2 Data**).

### 3.9 Ellagic acid is the most active ingredient from HCAE

Dietary polyphenols are known to exert their anti-cancer activity by interfering with membrane receptors, disrupting cellular signaling cascades, enzymes in basic metabolic processes, and other targets which basically holds up the cellular machinery [64]. Many crude extracts and their active ingredient have been shown to have anti-cancer action in synergistic and individualistic manner [65]. Often, it is the presence of a particular compound that is responsible for the activity of the crude extract. Thus, a deductive approach was employed to elucidate the polyphenols from HCAE which when removed will have a greater impact on the overall anti-cancer activity of the crude water extract (**Fig 8A**). Using the compositional make-up of 13 phytochemicals obtained from gradient HPLC chromatogram, different mixtures of the phytochemicals were made. The Complete Mixture (CM: mixture of all 13 identified compounds) did not show much variation in its anti-cancer activity (80.4 ± 5.5 μg/ml) when compared to its crude counterpart HCAE (70.41 ± 6.35 μg/ml). Further, the cellular viabilities ($IC_{50}$ values) of different mixtures (C1′-C13′) against the colon cancer cell line DLD1 revealed that subtraction of some polyphenols from the CM were more significant than others in terms of their anti-cancer activity. For instance, subtractions of individual polyphenols from the complete mixture such as C1′, C5′-C8′, C10′-C12′ observed a decrease in $IC_{50}$ values in the range of 0.99% to 19% that correlates to the gain of anti-cancer activity of HCAE (**Fig 8B**). This increase in anti-cancer activity may be attributed to their antagonistic action. Their antagonistic action may be (a) because of these phytochemicals competing for the same protein which is crucial in pathway of killing cancer cells or (b) may be because of formation of a complex polyphenol mixture which are unable to perform their individual activities [66, 67]. On the other hand, C3′, C9′ and C13′ which corresponds to subtraction of gallic acid, corilagin and ellagic acid respectively, observed an increase in the $IC_{50}$ values that signifies the loss of anti-cancer activity of HCAE. In C3′, C9′ and C13′, an increase in $IC_{50}$ values by 24.2%, 12.7% and 36.3% was observed, thus making ellagic acid as the most crucial polyphenol in the phytochemical makeup of the crude HCAE extract. Cancer cells when treated with ellagic acid exhibited loss of cellular viability and distortions in cellular morphology. Ellagic acid was found to exert anti-cancer activity in a dose dependent manner on colorectal cancer cell lines HCT-116, DLD1 & HT-29 with $IC_{50}$s of 10.08 ± 3.46 μg/ml, 17.39 ± 4.89 μg/ml, 13.1 ± 4.81 μg/ml respectively (**Fig 8C & 8D**). Ellagic acid is a well-known polyphenol which is known for its various activities that include antioxidant, neuroprotective, hepatoprotective etc., along with anti-cancer potential [63, 68].

### 3.10 Ellagic acid perturbs the cell cycle in cancer cells

Cancer cells treated with ellagic acid is exhibiting cellular stress with the appearance of distortion of cellular morphology. The stress linked signalling is associated with conservation of energy production and adaptation of cell with pro-survival strategies [69]. It in-turn disturbs many basic cellular functions including cell-cycle through reduction in production of various regulatory cyclins. It is well known that a mammalian cell undergoes various stages of cell growth and division [70]. The four phases present during cellular division are G1, S, G2, and M phases. The effect of ellagic acid on cell cycle progression in different colon cancer cell lines

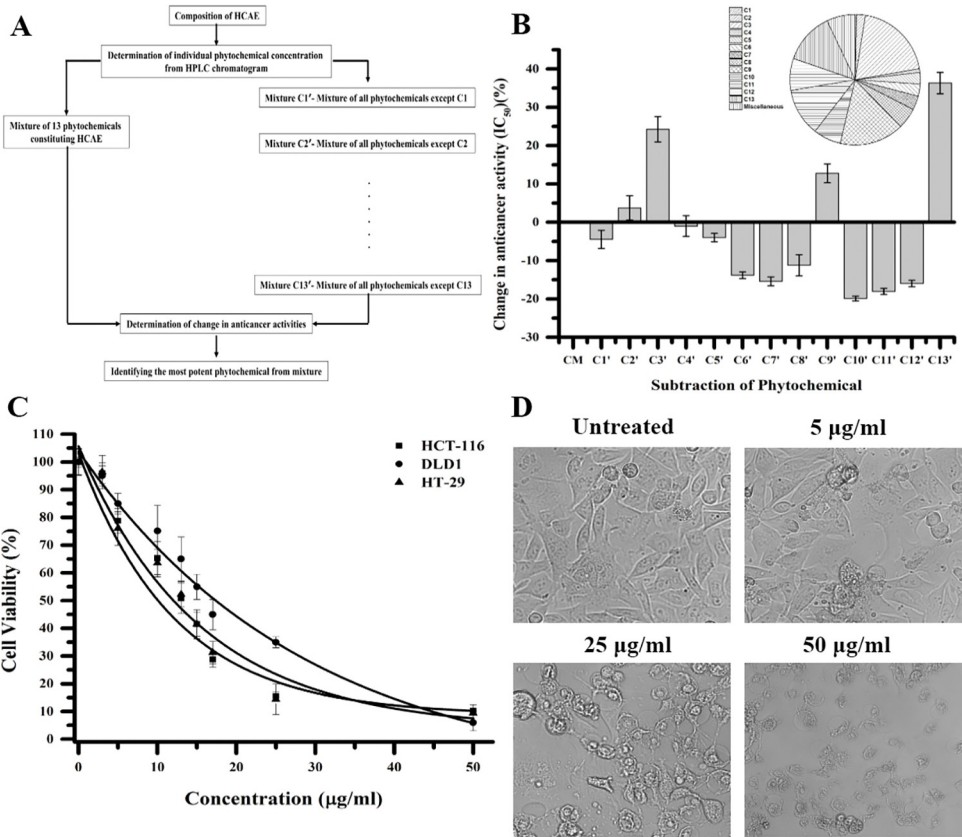

**Fig 8. Ellagic acid is the active ingredient responsible for anti-cancer activity of HCAE. (A)** Schematic of the deductive approach to discover the most active ingredient in HCAE. **(B)** The variation in anti-cancer activity after removal of compounds from HCAE. The term "CM" stands for the complete mixture, which is made up of 13 identified compounds and is created in accordance with the composition of the gradient HPLC chromatogram. **(C)** Colon Cancer cells (DLD1) were treated with different concentration of ellagic acid (0–50 μg/ml) and the cell viability was measured by MTT assay. The MTT curve of ellagic acid is presented with an $IC_{50}$ of 17.39± 4.89 μg/ml after treatment of 48 hours. **(D)** DLD1 cells were examined under a microscope, and pictures were captured from various fields using the Cytell cell imaging system (GE Healthcare). Representative images are shown. (**Note**: C1′-Mixture except shikimic acid, C2′-Mixture except chebulic acid, C3′-Mixture-except gallic acid, C4′-Mixture except 5-hydroxymethylfurfural, C5′-Mixture except protocatechuic acid, C6′-Mixture-except 4-*O*-galloyl shikimic acid, C7′-Mixture except 5-*O*-galloyl shikimic acid, C8′-Mixture except methyl gallate, C9′-Mixture-except corilagin, C10′-Mixture except 1,2,6-Tri-*O*-galloyl-β-D glucose, C11′-Mixture except chebulagic acid, C12′-Mixture-except chebulinic acid, C13′-Mixture except ellagic acid).

was assessed by flow cytometry analysis after staining the cells with propidium iodide (**Fig 9A**). The 2-D chromatogram was analysed using FCS Express software. Before treatment, the population of cells in S-phase were 17 ± 3.2%, 18.4 ± 4.1%, and 15.1 ± 6.9% for HCT-116, DLD1, and HT-29 cells respectively. Upon treatment with ellagic acid for 24h, the cell population in S-phase were 40.9 ± 6.1%, 30.8 ± 5.2%, and 31.9 ± 4.9%. The flow cytometry analysis suggests that there is an increase in the population of S-phase cells and cancer cells does not cross S-phase. Further, the protein expression levels of cyclin A2 and cyclin D1 were found to be down-regulated upon treatment with ellagic acid. Although cyclin A2 levels decreased after 24 hours of treatment, there were no significant changes in cyclin D1 expression levels, indicating that cells were arrested primarily in S-phase after 24 hours of treatment. However, after 48 hours of ellagic acid treatment, both cyclins (A2 and D1) were shown to have reduced expression levels in a dose dependent manner.

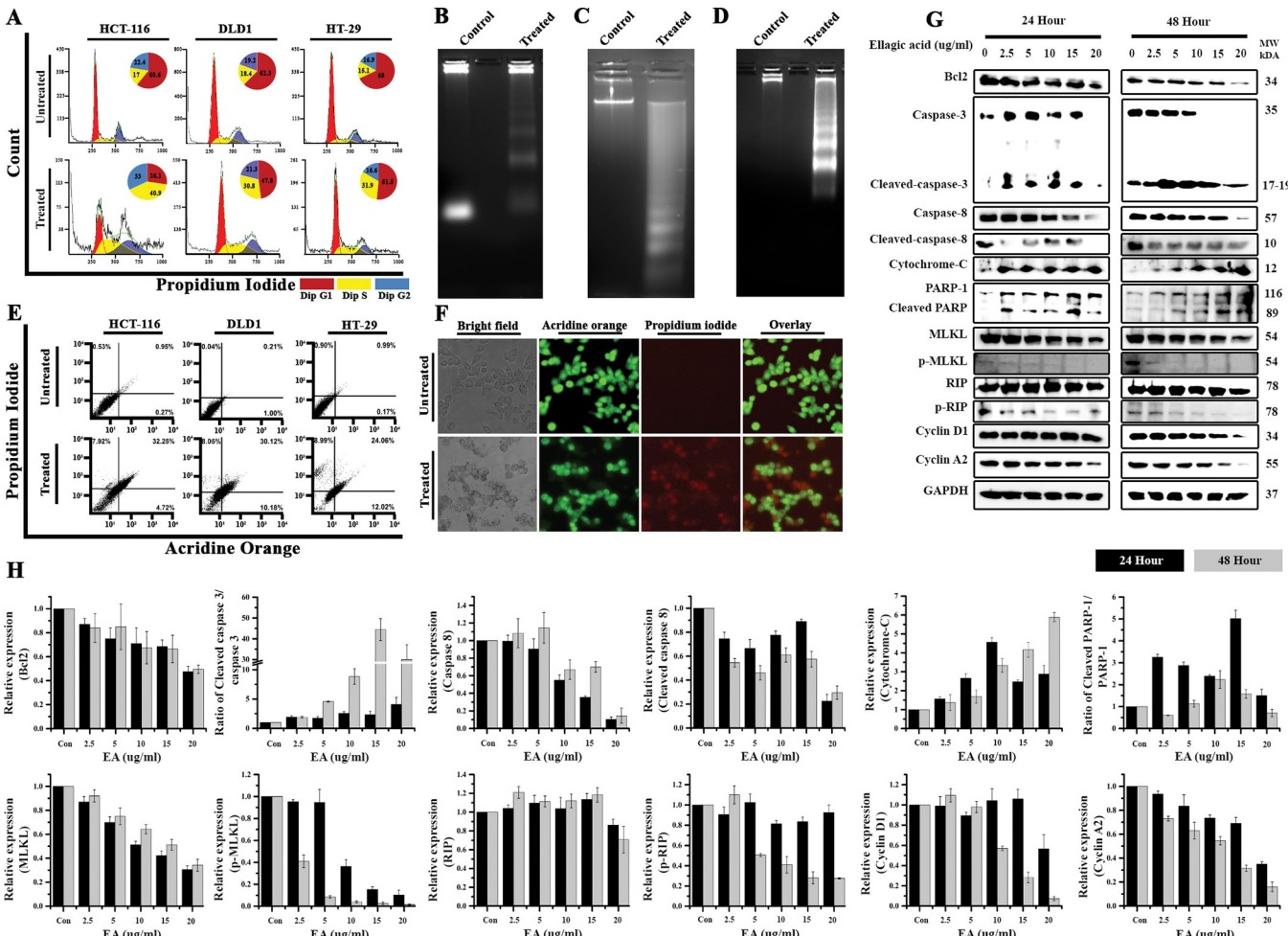

**Fig 9. Ellagic acid causes cell death in cancer cells by apoptosis. (A) Ellagic acid causes perturbations in cell cycle progression in cancer cells.** Ellagic acid treated colon cancer cell lines at their respective $IC_{50}$ concentrations for 24 hours were stained with propidium iodide and was subjected to analysis in FL-2A channel in BD FACS calibre. **Ellagic acid caused degradation of genomic DNA.** HCT-116 (**B**), DLD1 (**C**), and HT29 (**D**) cells were treated with ellagic acid shows the fragmentation of genomic DNA in the treated cells whereas the genomic DNA is intact in untreated cells. **(E) Ellagic acid induces apoptosis in cancer cells.** Flow cytometry analysis was performed on dual stained HCT-116, DLD-1 and HT-29 cells by BD FACS Caliber upon treatment with Haritaki churna aqueous extract for 48 hours with their respective $IC_{50}$ concentrations (10.08 μg/ml, 17.39 μg/ml and 13.1 μg/ml respectively). **(F)** Representative Images of Untreated and treated cancer cells from panel E. **(G) Ellagic acid induced death of colorectal cancer cells (DLD1) was caused by intrinsic pathway of apoptosis**. DLD1 cells were treated with various concentrations (2.5, 5, 10, 15, 20 μg/ml) of ellagic acids for 24 and 48 hours of treatment. Western blotting was performed after harvesting the proteins using RIPA lysis buffer. Western blot images of several proteins related to apoptosis, necroptosis, cell cycle regulation are shown. **(H)** Relative expression levels of proteins related to apoptosis, necroptosis, and cell cycle regulation.

## 3.11 Ellagic acid induces mitochondrial apoptotic pathway in cancer cells

The colorectal cancer cells in stress due to treatment with ellagic acid can induce cell death either by undergoing apoptosis, necrosis, autophagy or necroptosis. As caspase dependent apoptosis is the most common form of programmed cell death (PCD), we first checked DNA fragmentation/laddering which is a hallmark of apoptosis. Treatment of cells with ellagic acid resulted in DNA fragments that was detected by DNA laddering assay (**Fig 9B–9D**). It was further confirmed by dual staining of treated cells by acridine orange and propidium iodide is done to differentiate the live healthy cells from the apoptotic cells. Untreated cells were shown to have prominently healthy cells ranging from 94–99% in different cell lines (**Fig 9E**). The cells treated with ellagic acid were shown to have changed in the distribution of cells. The early

apoptotic was seen in a range of 5–12% compared to 0–1% in the untreated cells. There was also an increase in the late apoptotic cells which were seen in a range of 24–32% compared to 0–1% in the untreated cells. The FACS analysis was also confirmed by fluorescence imaging of the dually stained cells (**Fig 9F**). Further, immunoblotting of caspase 3 and its cleaved form obtained from DLD1 cells treated ellagic acid at 24h and 48 h revealed increased expression of cleaved caspase 3 (17–19 kDa) in a dose dependent manner (**Fig 9G & 9H**). Caspase 3 can be cleaved by two main pathways, the mitochondrial pathway via release of cytochrome-c or death receptor pathway via caspase 8 activation [71]. An increase in cytochrome-c expression was seen after 24 and 48 hours of ellagic acid treatment, indicating that the killing is caused via the mitochondrial apoptosis pathway. The formation of cleaved caspase 3 resulted in the cleavage of PARP-1 (poly (ADP-ribose) polymerase-1) to form cleaved PARP-1, an 89 kDa cleaved fragment indicating onset of apoptosis upon treatment with ellagic acid. The cells undergoing DNA damage/fragmentation or induction of ROS can lead to an over expression of pro-apoptotic proteins and simultaneously decrease anti-apoptotic proteins such as Bcl2 [72]. Hence, treatment with ellagic acid also down-regulated Bcl2. Additionally, the bioactive polyphenol was shown to downregulate caspase 8 and cleaved caspase 8 upon treatment, ruling out the possibility of killing cancer cells via classical death receptor pathway including TNFR1, TNFR2, and Fas receptors. Necroptosis is a type of controlled cell death caused by the activation of death receptors and the downregulation of caspase 8 [73]. The hallmark proteins of necroptosis, RIP (Receptor interacting protein kinase) and MLKL (Mixed lineage kinase domain-like), as well as their phosphorylated versions, p-RIP and p-MLKL, were shown to be down-regulated by ellagic acid treatment demonstrating no evidence of necroptosis (**Fig 9G & 9H**) (S1 Raw images).

The current study is aimed at exploring the anti-cancer properties and the bio-active molecules responsible from Haritaki churna. Ayurvedic formulations are of different types [1]. Based on the number of herbs used it can be either a single herb formulation or polyherbal formulation. Apart from this, they are also characterized based on their method of preparation such as ark, avaleh, churna, etc [74]. The search for new compounds from natural resources has gained popularity in the past. The discovery of drugs such as taxol, vincristine, etc., has motivated researchers to look for new drugs that are effective, economic, and have fewer side effects. The eminence of natural compounds depends on the careful selection of plants, their taxonomical data, and even information from ancient medicine [75]. There are a lot of plants available in ayurvedic medicine which is used in preparing different types of formulations. One of the most famous is Haritaki which is used in many formulations. Haritaki churna (prepared from plants of *Terminalia chebula* Retz.) is mainly used for treating gastro-intestinal disorders such as constipation, gastric ulcers, etc. Haritaki is used in manufacturing a single herb formulation consisting only of its principal plant or is also used to prepare polyherbal formulation such as Arkeshwara ras, Panchakola, Sapthaparna Rasagenthi Leyham Triphala [10–12, 19, 76–80]. There have been no studies reported on the anti-cancer activity of Haritaki (*Terminalia chebula* Retz.) as an ayurvedic formulation. However, there are many studies in which the researchers have shown the anti-cancer activity on the extracts of *Terminalia chebula* fruit. The study done by Saleem et al used the 70% methanolic extracts of *Terminalia chebula* Retz. and attributed the anti-cancer activity of their extract to the presence of chebulinic acid, tannic acid, and ellagic acid [61]. This complies with our results as we have shown that gallic acid, ellagic acid, and chebulinic acid to be among the major polyphenols present in our ayurvedic formulation. Similar types of studies were reported on the alcoholic extracts of *Terminalia chebula*. All three studies attributed their anti-cancer activity on different types of cancer cell lines to the presence of polyphenols in their extracts [49, 81, 82]. Haritaki churna aqueous extract

(HCAE) has anti-cancer properties where it is shown that HCAE helps in reducing cellular viability in a dose-dependent manner.

The polyphenols are known to have anti-oxidant properties and can exert chemo preventive effects towards different cancers. It does so by almost involving the whole spectrum of cell machinery such as inhibiting DNA synthesis, production of ROS, modulating cell survival pathways, cell cycle arrest, interference with membrane receptors, interaction with enzymes involved in metastasis & tumour promotion, etc. [83, 84]. The anti-cancer activity of HCAE investigated in this study in essence is due to polyphenols, with ellagic acid being the most potent one. The cell viability assays conducted on colorectal cancer cells suggests that ellagic acid unequivocally decreases their growth at lower concentrations and kills them at higher concentrations. These promising results are in agreement with the previously published results in which ellagic acid is shown to have effect on numerous cancer cells [62, 63, 85–87]. Further, ellagic acid was observed to have induced fragmentation of DNA which is a hallmark of apoptosis. Release of cytochrome-c from mitochondria, increased expression of cleaved caspase 3 and reduced expressions of caspase 8 and Bcl2 were observed indicating ellagic acid induced mitochondrial pathway of apoptosis. Consistent with our study, various apoptosis related events such as down regulation of Bcl-XL, cytochrome-c release from mitochondria, activation of caspase-9 and increase in expression of cleaved caspase-3 have occurred in Caco2 cells upon treatment with ellagic acid [88]. Several other colorectal cancer cells such as HCT-15, HCT-116, SW480, LoVo, HT-29 have observed similar fate when treated with ellagic acid [63, 85, 89]. Ellagic acid induced apoptosis is also linked with decreased levels of matrix metalloproteases (MMP2, MMP9) indicating suppression of metastatic potential of cancer cells. Accumulated evidence suggest that changes in cell cycle machinery can either induce or prevent apoptosis [90]. Cyclin D1 controls cancer cells' ability to adhere, invade, and metastasize in addition to its role in controlling the cell cycle [91]. Cyclin D1 overexpression and aberrations in wnt/β-catenin pathway is strongly linked to colorectal malignancies [92], and targeting this route with ellagic acid might be beneficial in the treatment of CRCs. Ellagic acid in this study was found to arrest cells specifically in S-phase in different colorectal cancer cell lines which correlated with reduced expression levels of cyclin A2 and cyclin D1. Our findings are in coherence with already published literature in which ellagic acid is found to promote cell cycle arrest and induce apoptosis by modulating PI3K/Akt pathway in colorectal cancer cells [93]. In a similar manner, ellagic acid was able to halt cell cycle in G0/G1-phase in MCF-7 (breast cancer cells) which was a result of the polyphenol modulating TGF-β/smads pathway [94]. Transcriptional profiling of ellagic acid treated Caco2 cells revealed changes in expression levels of cell cycle genes such as CCNB, CCNB1IP1 when there was arrest in S and G2/M phases of cell cycle [95]. This preliminary study sheds light on the ability of ellagic acid on interrupting basic cell machinery such as cell survival pathways, apoptosis regulation, cell proliferation pathways, etc.

## 4 Conclusions

Ayurvedic medicine makes use of many herbs and plants. One such plant is Haritaki (*Terminalia chebula* Retz.) which is considered one of the most important medicinal plants in ayurveda. Our study discovered that Haritaki churna is a potent anti-cancer ayurvedic formulation, and it has been noticed that Haritaki churna has been consistent in displaying anti-cancer action across different cancer cells, and it is stable under simulated gastric and intestinal circumstances. The loss of cellular viability of cancer cells was attributed to the presence of the high amount of polyphenols present in the HCAE. Gallic acid, methyl gallate, corilagin, chebulinic acid, and ellagic acid were the compounds contributing to the anti-cancer

activity of HCAE. Using a deductive method, it was discovered that ellagic acid was the most active compound in HCAE. Further, ellagic acid was also shown to induce apoptosis and arrest cell cycle which suggests its involvement in interruption of basic cellular machinery related to DNA synthesis and regulation of apoptosis. The increase in cytochrome-c expression, cleavage of caspase 3, and its target PARP-1 in a dose and time dependent manner shows that ellagic acid triggers cell death via the mitochondrial pathway of apoptosis, which is a feature of various chemotherapeutic treatments.

## Supporting information

**S1 Data. NMR and LC-MS data for compounds isolated from HCAE.**
(DOCX)

**S2 Data. Purity of compounds isolated from HCAE and quantification of GA, MG and EA in HCAE using standard curves.**
(DOCX)

**S3 Data. Cell viability assays for fractions and compounds isolated from HCAE against colorectal cancer (DLD1, HCT-116 and HT-29) cell lines.**
(DOCX)

**S1 Raw images. Raw images of western blots.**
(PDF)

## Acknowledgments

The authors acknowledge the technical support from central instrument facility "(CIF) of IIT Guwahati.

## Author Contributions

**Conceptualization:** Vishal Trivedi.

**Data curation:** Md Rafi Uz Zama Khan.

**Investigation:** Vishal Trivedi.

**Methodology:** Md Rafi Uz Zama Khan, Vishal Trivedi.

**Project administration:** Vishal Trivedi.

**Supervision:** Emiko Yanase, Vishal Trivedi.

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
