## [Decision Letter · Decision Letter 0]

8 Feb 2023

PONE-D-22-33541Extraction, phytochemical characterization and anti-cancer mechanism of Haritaki Churna: An ayurvedic formulation.PLOS ONE

Dear Dr. Trivedi,

Thank you for submitting your manuscript to PLOS ONE. After careful consideration, we feel that it has merit but does not fully meet PLOS ONE’s publication criteria as it currently stands. Therefore, we invite you to submit a revised version of the manuscript that addresses the points raised during the review process.

We look forward to receiving your revised manuscript.

Kind regards,

Chun-Hua Wang

Academic Editor

PLOS ONE

Journal Requirements:

2. We note that this submission includes NMR spectroscopy data. We would recommend that you include the following information in your methods section or as Supporting Information files:

a) The make/source of the NMR instrument used in your study, as well as the magnetic field strength. For each individual experiment, please also list: the nucleus being measured; the sample concentration; the solvent in which the sample is dissolved and if solvent signal suppression was used; the reference standard and the temperature.

b) A list of the chemical shifts for all compounds characterised by NMR spectroscopy, specifying, where relevant: the chemical shift (δ), the multiplicity and the coupling constants (in Hz), for the appropriate nuclei used for assignment.

c)The full integrated NMR spectrum, clearly labelled with the compound name and chemical structure.

We also strongly encourage authors to provide primary NMR data files, in particular for new compounds which have not been characterised in the existing literature. Authors should provide the acquisition data, FID files and processing parameters for each experiment, clearly labelled with the compound name and identifier, as well as a structure file for each provided dataset. See our list of recommended repositories here: https://journals.plos.org/plosone/s/recommended-repositories

"This work was supported by the Department of Science and Technology (DST-SERB) grant to V.T. MR acknowledges the financial support in the form of a fellowship from Indian Institute of Technology- Guwahati, Assam, India."

7. PLOS requires an ORCID iD for the corresponding author in Editorial Manager on papers submitted after December 6th, 2016. Please ensure that you have an ORCID iD and that it is validated in Editorial Manager. To do this, go to ‘Update my Information’ (in the upper left-hand corner of the main menu), and click on the Fetch/Validate link next to the ORCID field. This will take you to the ORCID site and allow you to create a new iD or authenticate a pre-existing iD in Editorial Manager. Please see the following video for instructions on linking an ORCID iD to your Editorial Manager account: https://www.youtube.com/watch?v=_xcclfuvtxQ

Reviewers' comments:

Reviewer's Responses to Questions

**Comments to the Author**

1. Is the manuscript technically sound, and do the data support the conclusions?

Reviewer #1: Yes

Reviewer #2: Yes

Reviewer #3: Partly

Reviewer #4: Yes

Reviewer #5: Partly

2. Has the statistical analysis been performed appropriately and rigorously? 

Reviewer #1: Yes

Reviewer #2: Yes

Reviewer #3: No

Reviewer #4: Yes

Reviewer #5: Yes

3. Have the authors made all data underlying the findings in their manuscript fully available?

Reviewer #1: Yes

Reviewer #2: Yes

Reviewer #3: No

Reviewer #4: Yes

Reviewer #5: Yes

4. Is the manuscript presented in an intelligible fashion and written in standard English?

Reviewer #1: No

Reviewer #2: No

Reviewer #3: Yes

Reviewer #4: Yes

Reviewer #5: No

5. Review Comments to the Author

Reviewer #1: The authors could improve the overall language of this manuscript to increase readability.

>Introduction: The introduction section of this study fails to relate with the previous study accurately. Rewrite this section and relate why this study is novel and its importance.

>Conclusion: This section is not properly written. You are suggested to write the major findings and its significance in few sentences.

>Some figures are blurry. Include figures with correct resolution.

Reviewer #2: I read with interest the paper and I believe that authors have taken into account a topic worthy to be investigated.

A good match between in vitro investigation and chemical characterization is clearly highlighted.

In my review I want to underline the strenghts, but I need to raise some fundamental limits of the work.

Strenghts:

I appreciate how authors planned the work, aiming first to provide with a simple but well performed protocol that was able to obtain some practical (and thus sounding) preclinical results.

Indeed the fact that authors studied the actual preparation used in ayurveda (i.e. maybe water is not the best solvent to extract Terminalia, but it is the solvent that practioners use), they considered the gastric stability, the effectiveness of phytocomplex, fractions and single compounds and tested samples in cancer and normal cell lines are all together basic good experimental practices in phytotherapy, too often underrated.

Limits:

I have to admit that this work opened many different interesting scientific questions, but many things remain unsolved.

The chemical characterization has been done but, due to the lack of reference standards, the quantification of ellagic acid as well other effective compounds is missing.

Authors should have completed the gastrointestonal stability simulation, by adding the simulation of the activity of bile salts and pancreatin at pH 6.8-7.2.

Finally, molecular mechanism were not investigated.

Please check carefully spelling errors.

Reviewer #3: The pictures in the article are not clear and the pixels do not meet the publication requirements.

The purity of the compound is not high.

Neither the fraction nor the compound activity is very strong.

Reviewer #4: This manuscript reports the extraction, phytochemical characterization, and anti-cancer mechanism of Haritaki Churna. The anti-cancer activity, safety, toxicity, stability analysis of Haritaki churna was tested, and the active ingredient significantly contributed into anti-cancer effect of HCAE was isolated and improved to played activity by degrading the genomic DNA and inducing the cellular viability,The overall logic of the paper is clear.Personally, the manuscript might require little revision prior to be considered to be acceptable or not due to that:

1. In figure 4, the retention time of B, and D need to be increased for easy observation about all compounds in the condition.

2. In figure 5, the structure of the compound is recommended to be beautified in the same format.

3. The accuracy of the mass spectrometry data in the article needs to be improved to four decimal places, and needs to be consistent, such as line 338 and 341.

4. In 3.5, please add the amount of each fraction separated.

Reviewer #5: The current study reported that the ayurvedic formulation, Haritaki churna (HC), has anti-cancer activity in different cancer cell lines and potential mechanisms. The authors checked the bioavailability of HC and it’s activity against different cancer cell lines. They further explored the active pharmaceutical ingredient in HC, tested the contributions of those phytochemicals, and found that Ellagic acid significantly contributes to the anti-cancer effect of HCAE. However, the present form can’t be accepted for publication.

1. The manuscript needs to be polished the English grammar or spelling by a native speaker.

Inconsistent hyphenation throughout the manuscript. Like anti-cancer vs anticancer, gastro-intestinal vs gastrointestinal, etc. Both ways are acceptable, but you have to keep consistent.

American English and British English are mixed used.

Grammar and punctuation mistakes everywhere

2. The last decade has seen remarkable progress in cancer research, with a significant update to many theories. But most of the citations here have been published for more than ten years.

The format of references is seriously inconsistent. For instance, ref 37 is in a different form from ref 36.

3. All experiment data need to have supported tables or figures, for section 3.1, no tables or figures are cited.

4. Section 3.3, the authors stated, “HC aqueous extract had not affected the cellular viability of HEK-293 cells up to concentration of 300 µg/ml. In addition, it is not affecting the cellular morphology as well (Figure 2B).”.

However, we can see significant changes in morphology in HEK293 in Figure 2B. And there is no discussion for these changes.

5. Figure 5, since there are different subfigures, please cite them separately according to the results.

6. Section 3.9, “This increase in anticancer activity may be attributed to their antagonistic action. Their antagonistic action may be (a) because of these phytochemicals competing for the same protein which is crucial in pathway of killing cancer cells or (b) may be because of formation of a complex polyphenol mixture which are unable to perform their individual activities. “

References to support this discussion are missing.

7. Section 3.10. Three independent repeated experiments are needed to get the mean ± SD/ SEM and reliable results. While the results here seem only come from a one-off experiment.

8. Current data is not strong enough to make the conclusion that “ellagic acid is capable of arresting cell cycle progression in S-phase and induction of apoptosis in colorectal cancer cells”

More experiment such as Western Bloting to check the cell cyle stage-dependant makers is needed, such as Cyclin A, Cyclin B…

The same for apoptosis, more experimental evidence is needed.

6. PLOS authors have the option to publish the peer review history of their article (what does this mean?). If published, this will include your full peer review and any attached files.

Reviewer #1: No

Reviewer #2: **Yes: **Marco Biagi

Reviewer #3: No

Reviewer #4: No

Reviewer #5: No

---

## [Author Response · Author response to Decision Letter 0]

5 Apr 2023

From 11.03.2023 

Dr. Vishal Trivedi 

Professor, 

Department of Biosciences and Bioengineering 

Indian Institute of Technology-Guwahati 

Guwahati-Assam India-781039 

To 

The Editor, 

Plos One

Dear Editor, 

Thank you for giving us the opportunity to submit a revised draft of our manuscript titled “Extraction, phytochemical characterization and anti-cancer mechanism of Haritaki Churna: An ayurvedic formulation” to Plos One journal We appreciate the time and work you and the reviewers put into giving useful criticism on the paper. We have been able to incorporate changes to reflect most of the suggestions provided by the reviewers. We have highlighted the changes within the manuscript.

The revised manuscript has sufficient merit to be published in your Journal. We have no conflict of interest in any part of this article and none of the material has been published or is under consideration elsewhere. 

All authors have contributed to, seen, and approved the final, submitted version of the manuscript. 

With best regards, 

Vishal Trivedi

 

Response to reviewer

Reviewer 1: 

Comment 1: The authors could improve the overall language of this manuscript to increase readability.

Response: English is edited, corrected by online English editing software Grammarly and another colleague proof read the manuscript. Language is of journal’s desired standard. 

Comment 2: Introduction: The introduction section of this study fails to relate with the previous study accurately. Rewrite this section and relate why this study is novel and its importance.

Response to comment 2: The introduction section has been modified as per the requirement. (L56-L80)

Comment 3: Conclusion: This section is not properly written. You are suggested to write the major findings and its significance in few sentences.

Response to comment 3: As suggested, The conclusion section has been modified as per the requirement. (L626-639)

Comment 4: Some figures are blurry. Include figures with correct resolution

Response to comment 4: The figures re-submitted to journal are of publication quality with high resolution of 300 dpi. All the figures have been prepared again as per journal requirements. 

Reviewer 2:

Comment 1: The chemical characterization has been done but, due to the lack of reference standards, the quantification of ellagic acid as well other effective compounds is missing.

Response to comment 1: Quantification of gallic acid, methyl gallate and ellagic was done using the respective standards. The standard curves were drawn after spiking different concentrations of GA, MG and EA in a gradient HPLC chromatogram. The calibration curve of different phytochemical was used to quantify. This data is included in supplementary data-2. The write up of quantification of GA, MG and EA are included in main manuscript from Line 477-481.

Comment 2: Authors should have completed the gastrointestinal stability simulation, by adding the simulation of the activity of bile salts and pancreatin at pH 6.8-7.2.

Response to comment 2: 

• In order to test the efficacy of HC and HCAE in simulated intestinal fluids (SIF), two types of dissolution media were used that could mimic the condition in fasted (FaSSIF) and fed state (FeSSIF). These dissolution media primarily contain bile salts and pancreatin at pH 6.5 and pH 5.8

• The recipe to prepare FaSSIF and FeSSIF have been included in Materials and methods section as section 2.6.2 and 2.6.3

• HC was extracted in FaSSIF and FeSSIF and no significant changes in the compositional make up of the extracts when compared to its aqueous extract. Further, HCAE was incubated in FaSSIF and FeSSIF from 0-6 hours at a time interval of 2 hours. A decrease in reduction in intensity of compound 11, 12 (chebulagic acid and chebulinic acid) and an increase in intensity of compound 13 (ellagic acid) was observed with progression of incubation periods. The increase in intensity of ellagic acid in HCAE incubated with FaSSIF and FeSSIF showed reduction in IC50 values when tested on DLD1 cancer cells for 48 hours of treatment. 

• The write up explaining the result have been discussed in section 3.7 from Line 424-440

• A new figure (Figure 7) have been included in manuscript to explain the results of effect of simulated intestinal conditions on HCAE

Comment 3: Finally, molecular mechanism was not investigated.

Response to comment 3: 

• Molecular mechanism explaining the death of colorectal cancer cells is discussed in section 3.11. Further, the in-depth mechanism of killing of DLD1 cancer cells upon treatment with ellagic acid was discussed with respect to up-regulation and down-regulation of apoptotic, necroptotic and cell signaling proteins through western blotting. 

• The write up of results of western blotting experiments are discussed in section 3.10 & 3.11 from Line 521-563

• Two new panels (9G & 9H) have been added in Figure 9

Comment 4: Please check carefully spelling errors.

Response to comment 4: Spelling errors have been corrected wherever necessary. English is edited, corrected by online English editing software Grammarly and another colleague proof read the manuscript. Language is of journal’s desired standard. 

Reviewer 3: 

Comment 1: The pictures in the article are not clear and the pixels do not meet the publication requirements.

Response to comment 1: The figures re-submitted to journal are of publication quality with high resolution of 300 dpi. All the figures have been prepared again as per journal requirements

Comment 2: The purity of the compound is not high.

Response: I agree with your observation as few minor peaks are appearing in HPLC chromatogram of isolated phytochemicals from HCAE (Supplementary data 2). The most of the phytochemicals isolated from HCAE and used in the current study are more than 95% pure. It is known that the isolated pure phytochemicals from plant are prone to degradation, modification (oxidation) and appearance of additional peaks on the HPLC chromatogram could be due to these reasons. The compound present in minor peaks is very less and may not cause any significant impact on the anti-cancer activity reported by us. 

Comment 3: Neither the fraction nor the compound activity is very strong.

Response to comment 3: The individual fractions anti-cancer activities has been discussed section 3.8. All the fractions are not active, but fraction 2 and fraction 5 show greater activity than the aqueous extract. Since it was a deductive approach, we were able to rule out certain fractions that were not as good as fractions 2 and 5. Further, individual anti-cancer activities of each compound have also been reported in the manuscript which correlate with the previously published literature.

Reviewer 4:

Comment 1: In figure 4, the retention time of B, and D need to be increased for easy observation about all compounds in the condition.

Response to comment 1: Figure 4 have been updated. The retentions times of all the fractions now show from 0 to 30 minutes

Comment 2: In figure 5, the structure of the compound is recommended to be beautified in the same format.

Response to comment 2: Figure 5 have been beautified as per requirement. 

Comment 3: The accuracy of the mass spectrometry data in the article needs to be improved to four decimal places, and needs to be consistent, such as line 338 and 341.

Response to comment 3: The accuracy of the mass spectrometer data has been updated to four decimals in section 2.13 and section 3.6

Comment 4: In 3.5, please add the amount of each fraction separated.

Response to comment 4: The amount of each fraction separated is added in section 3.5 from Line 350-352

Reviewer 5:

Comment 1: The manuscript needs to be polished the English grammar or spelling by a native speaker.

Inconsistent hyphenation throughout the manuscript. Like anti-cancer vs anticancer, gastro-intestinal vs gastrointestinal, etc. Both ways are acceptable, but you have to keep consistent.

American English and British English are mixed used.

Response to comment 1: English grammatical mistakes were corrected through out the manuscript and consistency of single language system is maintained. 

Comment 2: The last decade has seen remarkable progress in cancer research, with a significant update to many theories. But most of the citations here have been published for more than ten years.

Response to comment 2: The citations with reference to cancer research have been updated wherever applicable. New references have been added and old references have been removed.

Comment 3: All experiment data need to have supported tables or figures, for section 3.1, no tables or figures are cited.

Response to comment 3: Table 1 has been cited in section 3.1 and has been updated in line 297

Comment 4: Section 3.3, the authors stated, “HC aqueous extract had not affected the cellular viability of HEK-293 cells up to concentration of 300 µg/ml. In addition, it is not affecting the cellular morphology as well (Figure 2B). However, we can see significant changes in morphology in HEK293 in Figure 2B. And there is no discussion for these changes.

Response to comment 4: “Blebbing in HEK-293 cells treated with higher concentration (500µg/ml) of HCAE was observed (Figure 2B). However, these protrusions have been found in HEK-293 cells at doses greater than those necessary to kill colorectal cancer cells”. The discussion pertaining to comment 4 have been updated in Line 328-330

Comment 5: Figure 5, since there are different subfigures, please cite them separately according to the results.

Response to comment 5: Subfigures of figure 5 have been cited next to the names of the compounds identified in section 3.6

Comment 6: Section 3.9, “This increase in anticancer activity may be attributed to their antagonistic action. Their antagonistic action may be (a) because of these phytochemicals competing for the same protein which is crucial in pathway of killing cancer cells or (b) may be because of formation of a complex polyphenol mixture which are unable to perform their individual activities. References to support this discussion are missing

Response to comment 6: Relevant reference supporting the statement is been added in the revised manuscript.

Comment 7: Section 3.10. Three independent repeated experiments are needed to get the mean ± SD/ SEM and reliable results. While the results here seem only come from a one-off experiment.

Response to comment 7: Standard deviation values for the experiment discussed in section 3.10 have been updated in Line 524-526 

Comment 8: Current data is not strong enough to make the conclusion that “ellagic acid is capable of arresting cell cycle progression in S-phase and induction of apoptosis in colorectal cancer cells” More experiment such as Western Blotting to check the cell cycle stage-dependent makers is needed, such as Cyclin A, Cyclin B…

The same for apoptosis, more experimental evidence is needed.

Response to comment 8: 

• Molecular mechanism explaining the death of colorectal cancer cells is discussed in section 3.11. Further, the in-depth mechanism of killing of DLD1 cancer cells upon treatment with ellagic acid was discussed with respect to up-regulation and down-regulation of apoptotic, necroptotic and cell signaling proteins through western blotting. 

• Western blotting of caspase 8, cleaved caspase 8, caspase 3, cleaved caspase 3, PARP-1, Bcl2, cytochrome-c, MLKL, p-MLKL, RIP, and p-RIP have been included in the results section which rule out extrinsic pathway of apoptosis and necroptosis happening upon treatment with ellagic acid at different concentration for 24 and 48 hours.

• Further, expression levels of cyclin A2 and cyclin D1 were recorded using western blotting. Cyclin A2 levels lowered in both time points, but cyclin D1 was predominantly lowered in 48 hours of treatment.

• The write up of results of western blotting experiments are discussed in section 3.10 & 3.11 from Line 521-563

• Two new panels (9G & 9H) have been added in Figure 9 showing the western blot results obtained. 

• The discussion and conclusion section have also been updated with reference to the new data included in the manuscript.

---

## [Decision Letter · Decision Letter 1]

12 May 2023

Extraction, phytochemical characterization and anti-cancer mechanism of Haritaki Churna: An ayurvedic formulation.

PONE-D-22-33541R1

Dear Dr. Trivedi,

We’re pleased to inform you that your manuscript has been judged scientifically suitable for publication and will be formally accepted for publication once it meets all outstanding technical requirements.

Kind regards,

Chun-Hua Wang

Academic Editor

PLOS ONE

Additional Editor Comments (optional):

Reviewers' comments:

Reviewer's Responses to Questions

**Comments to the Author**

1. If the authors have adequately addressed your comments raised in a previous round of review and you feel that this manuscript is now acceptable for publication, you may indicate that here to bypass the “Comments to the Author” section, enter your conflict of interest statement in the “Confidential to Editor” section, and submit your "Accept" recommendation.

Reviewer #1: All comments have been addressed

Reviewer #5: All comments have been addressed

2. Is the manuscript technically sound, and do the data support the conclusions?

Reviewer #1: Yes

Reviewer #5: Yes

3. Has the statistical analysis been performed appropriately and rigorously? 

Reviewer #1: Yes

Reviewer #5: Yes

4. Have the authors made all data underlying the findings in their manuscript fully available?

Reviewer #1: Yes

Reviewer #5: Yes

5. Is the manuscript presented in an intelligible fashion and written in standard English?

Reviewer #1: Yes

Reviewer #5: Yes

6. Review Comments to the Author

Reviewer #1: (No Response)

Reviewer #5: (No Response)

7. PLOS authors have the option to publish the peer review history of their article (what does this mean?). If published, this will include your full peer review and any attached files.

Reviewer #1: No

Reviewer #5: No

---

## [Editor Report · Acceptance letter]

22 May 2023

PONE-D-22-33541R1 

Extraction, phytochemical characterization and anti-cancer mechanism of Haritaki Churna: An ayurvedic formulation. 

Dear Dr. Trivedi:

I'm pleased to inform you that your manuscript has been deemed suitable for publication in PLOS ONE. Congratulations! Your manuscript is now with our production department. 

Kind regards, 

on behalf of

Dr. Chun-Hua Wang 

Academic Editor

PLOS ONE